# VERL-TOOL: TOWARDS HOLISTIC AGENTIC REINFORCEMENT LEARNING WITH TOOL USE

## ABSTRACT

Reinforcement Learning with Verifiable Rewards (RLVR) has demonstrated success in enhancing LLM reasoning capabilities, but remains limited to single-turn interactions without tool integration. While recent **A**gentic **R**einforcement **L**earning with **T**ool use (ARLT) approaches have emerged to address multi-turn tool interactions, existing works develop task-specific codebases that suffer from fragmentation, synchronous execution bottlenecks, and limited extensibility across domains. These inefficiencies hinder broader community adoption and algorithmic innovations. We introduce **VERL-TOOL**, a unified and modular framework that addresses these limitations through systematic design principles. **VERL-TOOL** provides four key contributions: **(1)** upstream alignment with VeRL ensuring compatibility and simplified maintenance, **(2)** unified tool management via standardized APIs supporting diverse modalities including code execution, search, SQL databases, and vision processing, **(3)** asynchronous roll-out execution achieving near $2\times$ speedup by eliminating synchronization bottlenecks, and **(4)** comprehensive evaluation demonstrating competitive performance across 6 ARLT domains. Our framework formalizes ARLT as multi-turn trajectories with multi-modal observation tokens (text/image/video), extending beyond single-turn RLVR paradigms. We train and evaluate models on mathematical reasoning, knowledge QA, SQL generation, visual reasoning, web search, and software engineering tasks, achieving results comparable to specialized systems while providing a unified training infrastructure. The modular plugin architecture enables rapid tool integration requiring only lightweight Python definitions, significantly reducing development overhead and providing a scalable foundation for tool-augmented RL research.

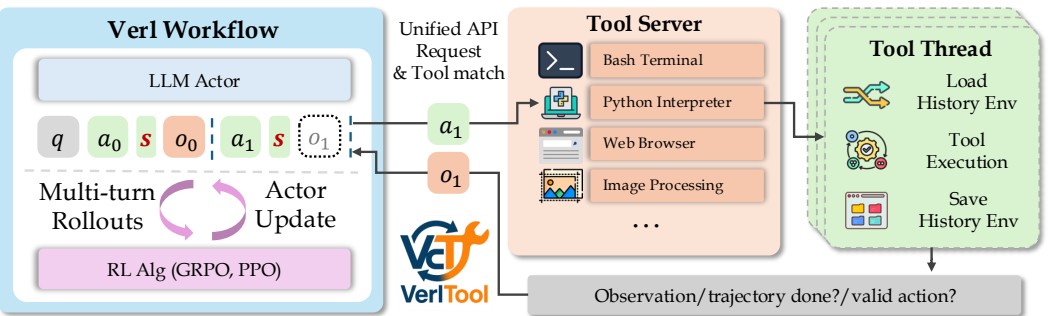

Figure 1: Overview of the **VERL-TOOL**, a modularized and efficient framework for the **A**gentic **R**einforcement **L**earning with **T**ool Use (ARLT) training paradigm, where the RL workflow and tool execution are fully disaggregated for both efficiency and extensibility.

## 1 INTRODUCTION

> *"We shape our tools, and thereafter our tools shape us."*     — Marshall McLuhan

Large language models (LLMs) such as OpenAI's O-series (Jaech et al., 2024) and DEEPSEEK-R1 (Guo et al., 2025) have recently achieved striking advances, surpassing top human performers in

challenging domains like mathematics (AIME) and programming (LIVECODEBENCH (Jain et al., 2024), CODEFORCES (Quan et al., 2025)). A central driver of this progress is the paradigm of *reinforcement learning with verifiable rewards* (RLVR), which strengthens long-context reasoning during training. Through RLVR, LLMs exhibit emergent cognitive behaviors such as reflection, backtracking, and multi-step reasoning.

Yet these systems remain constrained in a fundamental way: they are unable to interact with the external world. Current LLM reasoning unfolds in a closed, single-turn setting without environmental feedback, often leading to brittle behaviors such as overthinking (Chen et al., 2024a) or hallucination (Yao et al., 2025). Conceptually, these models resemble a "brain in a vat", locked into self-contained simulations without grounding in interactive or physical reality.

To overcome this isolation, a parallel line of work has explored augmenting LLMs with the ability to use tools. Systems such as TOOLFORMER (Schick et al., 2023) and OPENHANDS (Wang et al., 2024b) extend models with supervised training on synthetic tool-use data, enabling practical interaction with code interpreters, search engines, or APIs. However, these approaches primarily rely on imitation learning. They lack the agentic autonomy needed to learn directly from feedback and refine their behavior adaptively in open-ended environments.

Recent research begins to bridge this gap by combining tool use with RLVR, giving rise to a new paradigm we term **ARLT**—**A**gentic **R**einforcement **L**earning with **T**ool use. In ARLT, LLMs can actively engage with external tools such as code execution environments (Li et al., 2025c), search engines (Jin et al., 2025), image manipulators (Su et al., 2025), and domain-specific APIs (Feng et al., 2025). This interaction transforms training into a multi-turn, feedback-rich process that not only improves efficiency and reduces token usage but also fosters more robust agentic behaviors.

However, enabling ARLT poses significant challenges from a systems perspective. First, *rollout efficiency* becomes critical: multi-tool trajectories unfold asynchronously, with different tools producing results at varying speeds, demanding scalable asynchronous execution. Second, *tool management* remains fragmented: existing ARLT codebases are often tailored to specific tools, making it difficult to extend or reproduce results. Finally, *multimodal support* is still underdeveloped: while most RL frameworks focus narrowly on text, emerging multimodal reasoning agents (e.g., PIXEL-REASONER (Su et al., 2025)) require handling tool outputs that include images, videos, or other structured modalities in a unified design.

These barriers have slowed community progress, limiting reproducibility, extensibility, and algorithmic innovation. To address them, we introduce VERL-TOOL: an open-source, user-friendly, and efficient framework built on top of VERL (Sheng et al., 2024), designed explicitly for ARLT that supports both text and multimodal training. Unlike prior systems, VERL-TOOL enables multi-turn, stateful agentic training with tool use through four key contributions:

- **Upstream Alignment.** VERL-TOOL inherits VERL as a submodule, ensuring compatibility with upstream updates. This modular separation between RL training and agentic interaction simplifies maintenance and accelerates framework evolution.
- **Unified Tool Management.** We introduce a dedicated tool server with standardized interaction APIs, supporting diverse tools such as code execution, search, SQL/tabular reasoning, and vision utilities. Adding a new tool requires only a lightweight Python definition file, streamlining extensibility for both training and evaluation.
- **Asynchronous Rollouts.** By interacting with tool servers on a trajectory-by-trajectory basis rather than synchronously batch by batch, VERL-TOOL eliminates idle waiting time. This design yields over $2\times$ speedup during rollout execution.
- **Diverse ARLT Tasks.** We have implemented and tested VERL-TOOL on six ARLT tasks, including Math, Search Retrieval, SQL, Visual Reasoning, Web Browsing, and SWE-Bench, achieving competitive performance with previous baselines while trained in a unified framework. We also present common findings in the agentic RL setting across these tasks.

In summary, VERL-TOOL provides a principled, extensible, and efficient framework for ARLT, bridging the gap between isolated LLM reasoning and interactive agentic intelligence. By combining upstream-aligned RL infrastructure, unified tool integration, asynchronous execution, and diverse tasks, it paves the way for scalable research and practical deployment of LLMs as tool-using agents.

## 2 Related Work

### 2.1 Reinforcement Learning for Agentic Tool Use

The integration of reinforcement learning with tool use has emerged as a powerful paradigm for developing adaptive LLM agents. Early tool-calling approaches relied on prompt-based orchestration (Yao et al., 2022; Lu et al., 2023; Shen et al., 2023), building on Chain-of-Thought reasoning (Wei et al., 2022) and multi-agent frameworks for training-free tool invocation. While instruction-tuned models (Schick et al., 2023; Kong et al., 2023; Gou et al., 2023) learned structured calling patterns through supervised learning, they remained largely static and limited to single-turn interactions.

Recent work has demonstrated the advantages of reinforcement learning for tool use, enabling models to optimize their tool-calling policies based on execution outcomes and environmental feedback (Li et al., 2025c; Feng et al., 2025; Moshkov et al., 2025; Wang et al., 2025a). This paradigm, which we refer to as *Agentic Reinforcement Learning with Tool use (ARLT)*, extends beyond single-turn verification to support long-horizon, multi-turn interactions. Key characteristics of ARLT include: (1) credit assignment across sequential tool calls, (2) explicit handling of observation tokens from tool responses, and (3) integration with robust, failure-aware execution environments (Plaat et al., 2025; Ke et al., 2025).

This shift from static instruction-following to dynamic, feedback-driven learning has shown effectiveness across diverse domains, including mathematical reasoning with code execution, information retrieval, natural language to SQL generation, and visual reasoning tasks. These applications require agents to probe environments iteratively, adapt to tool feedback, and refine their strategies—capabilities that are difficult to achieve through purely supervised approaches.

### 2.2 Agentic RL Training Frameworks

Table 1: Tool support comparison across different frameworks (up to update until August 23, 2025). RAGEN and ROLL focus on the puzzle environments like bandit, which we did not list here.

| Framework | FAISS Search | Python Executor | Web Search | Bash Terminal | SQL Executor | Image Processing |
|---|---|---|---|---|---|---|
| OPENRLHF (Hu et al., 2024) | ✓ | ✓ | ✗ | ✗ | ✗ | ✗ |
| VERL (Sheng et al., 2024) | ✓ | ✓ | ✗ | ✗ | ✗ | ✗ |
| ROLL (Wang et al., 2025b) | ✗ | ✗ | ✗ | ✗ | ✗ | ✗ |
| RAGEN (Wang et al., 2025c) | ✗ | ✗ | ✗ | ✗ | ✗ | ✗ |
| SLIME (THUDM, 2024) | ✓ | ✗ | ✗ | ✗ | ✗ | ✗ |
| AREAL (Fu et al., 2025) | ✓ | ✗ | ✓ | ✗ | ✗ | ✗ |
| SKYRL (Cao et al., 2025) | ✗ | ✓ | ✗ | ✓ | ✓ | ✗ |
| **VERL-TOOL** (ours) | ✓ | ✓ | ✓ | ✓ | ✓ | ✓ |

The success of Reinforcement Learning from Verifier Rewards (RLVR) has motivated the development of various frameworks to support scalable RL training for language models. Established synchronous frameworks include OPENRLHF (Hu et al., 2024) and VERL (Sheng et al., 2024), which employ Ray-based distributed computing to manage training workflows. Additionally, fully asynchronous frameworks such as AREAL (Fu et al., 2025), ROLL (Wang et al., 2025b), and SLIME (THUDM, 2024) have emerged to address scalability challenges.

As shown in Table 1, existing frameworks exhibit varying degrees of tool support. Traditional RL frameworks such as OPENRLHF and VERL provide basic support for search and code execution tools but lack comprehensive multi-modal capabilities. ROLL focuses primarily on core RL training without extensive tool integration, while AREAL supports search functionality but has limited executor capabilities. SKYRL (Cao et al., 2025) offers broader tool support, including bash terminals and SQL executors, but requires complex containerized environments that introduce deployment overhead. The limited tool coverage in existing frameworks has led to the development of domain-specific systems (e.g., SEARCH-R1, PIXELREASONER, and TOOLRL) as task-specific extensions.

However, these implementations typically feature hard-coded tool integrations that limit their extensibility and adaptability to new domains. As evident from Table 1, there remains a need for frameworks that provide comprehensive, extensible support for diverse tool types while maintaining ease of deployment and development.

# 3 VERLTOOL FRAMEWORK

In this section, we formulate the conceptual foundation of the ARLT paradigm starting from the original RLVR setting. We then elaborate on how VERL-TOOL serves as a practical implementation on the server side for **A**gentic **R**einforcement **L**earning with **T**ool use (**ARLT**).

## 3.1 PRELIMINARIES

**RLVR**  Reinforcement learning with verifiable reward (RLVR) optimizes the language model using a predefined verifiable reward via the following objective:

$$\max_{\pi_\theta} \mathbb{E}_{x \sim \mathcal{D}, y \sim \pi_\theta(\cdot|x)} \left[ R_\phi(x, y) \right] - \beta \, \mathbb{D}_{\mathrm{KL}} \left[ \pi_\theta(y \mid x) \, \| \, \pi_{\mathrm{ref}}(y \mid x) \right], \tag{1}$$

where $\pi_\theta$ denotes the policy LLM, $\pi_{\mathrm{ref}}$ is the reference LLM, $R_\phi$ is the verifiable reward function, and $\mathbb{D}_{\mathrm{KL}}$ is the KL divergence. The input $x$ is drawn from the dataset $\mathcal{D}$, and $y$ is the corresponding single-turn output. A typical verifiable reward function is defined as:

$$R_\phi(x, y) = \begin{cases} 1 & \text{if match}(y, y_g) \\ -1 & \text{otherwise} \end{cases} \tag{2}$$

where $y_g$ is the ground-truth answer and $\mathrm{match}(\cdot, \cdot) \in \{1, 0\}$ is a verification function that determines whether the generated answer $y$ matches $y_g$. This function can be implemented using either rule-based approaches (Wang et al., 2024b) or model-based verifiers (Ma et al., 2025).

**GRPO** (Shao et al., 2024) is a widely adopted RL algorithm designed to optimize the objective in Equation 1. In the single-turn RL case, the trajectory is simply the LLM generation $\tau = \{y\}$. The GRPO objective is given by:

$$J_{\mathrm{GRPO}}(\theta) = \frac{1}{G} \sum_{i=1}^{G} \frac{1}{|\tau_i|} \sum_{t=1}^{|\tau_i|} \min \left[ r_{i,t}(\theta) \cdot \hat{A}_{i,t}, \, \mathrm{clip}\left( r_{i,t}(\theta), 1 - \epsilon, 1 + \epsilon \right) \cdot \hat{A}_{i,t} \right], \tag{3}$$

where $r_{i,t}(\theta)$ is the token-level importance ratio and $\hat{A}_{i,t}$ is the normalized advantage across all tokens:

$$r_{i,t}(\theta) = \frac{\pi_\theta(\tau_{i,(t)} | \tau_{i,<t})}{\pi_{\mathrm{old}}(\tau_{i,(t)} | \tau_{i,<t})}, \quad \hat{A}_{i,t} = \frac{R_\phi(x, y) - \mathrm{mean}(\{R_\phi(\tau_1), \ldots, R_\phi(\tau_G)\})}{\mathrm{std}(\{R_\phi(\tau_1), \ldots, R_\phi(\tau_G)\})}. \tag{4}$$

## 3.2 AGENTIC REINFORCEMENT LEARNING WITH TOOL USE

**ARLT.**  In the agentic RL setting, rollouts are *multi-turn* instead of single-turn, and the agent can *interact with tools* to receive external observations during the reasoning process. Thus, the trajectory can be written as $\tau = \{a_0, o_0, \ldots, a_{n-1}, o_{n-1}, a_n\}$, where $a_i$ denotes the LLM-generated action tokens and $o_i$ denotes the observation tokens returned by a tool call. Here, $n$ is the total number of interaction steps.

To determine whether an action $a_i$ invokes a specific tool, we assume that each $a_i$ (for $0 \le i < n$) ends with a stop token $s \in \mathbb{S}_k$, where $\mathbb{S}_k$ is the predefined set of stop tokens for tool $T_k \in \mathbb{T}$. For example, $\mathbb{S}_{\mathrm{CI}} = \{\texttt{```output}, \texttt{</python>}\}$ for a code interpreter tool, or $\mathbb{S}_{\mathrm{search}} = \{\texttt{</search>}\}$ for a search tool. The complete set of stop tokens is the union over all invoked tools: $\mathbb{S} = \bigcup_{k=1}^{|\mathbb{T}|} \mathbb{S}_k$.

The introduction of observation tokens $o_i$ makes ARLT fundamentally different from the agentic RL defined in RAGEN (Wang et al., 2025c), where the agent only receives scalar rewards through environmental interaction. Moreover, the observation tokens are off-policy with respect to the current LLM $\pi_\theta$ being optimized, which can destabilize training (Jin et al., 2025). Therefore, these tokens

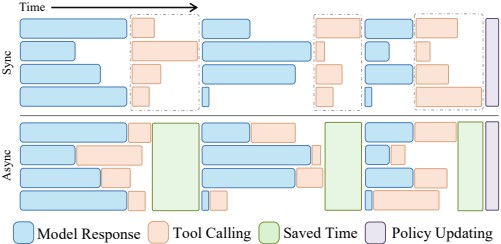

Figure 2: Visualization of the Async Rollout pipeline design and its effect in saving time.

Table 2: Performance comparison of Synchronous vs Asynchronous approaches. Experiments conducted on 8 H100 GPUs.

| Metrics | Math-TIR | SQL | DeepSearch |
|---|---|---|---|
| Turns | 4 | 5 | 5 |
| Sync (s) | 87 | 111 | 193 |
| Async (s) | 66 | 91 | 98 |
| **Speed Up** ($\times$) | **1.32** | **1.22** | **1.97** |

are typically masked out during policy optimization. Let $T_j$ be the token index of the first token in action segment $a_j$, then the GRPO loss for ARLT becomes:

$$J_{\text{GRPO-ARLT}}(\theta) = \frac{1}{G} \sum_{i=1}^{G} \frac{1}{\sum_{j=0}^{n} |a_j|} \sum_{j=0}^{n} \sum_{t=T_j}^{T_j + |a_j|} \min \left[ r_{i,t}(\theta) \cdot \hat{A}_{i,t}, \text{clip}\left(r_{i,t}(\theta), 1 - \epsilon, 1 + \epsilon\right) \cdot \hat{A}_{i,t} \right],$$

(5)

## 3.3 FRAMEWORK DESIGN

**Challenges.** Building a general RL training framework that supports various tools is inherently challenging due to the additional overhead introduced by tool interactions. First, prior ARLT works are designed around a single tool and tightly couple the tool interaction logic with the core RL training loop, making it difficult for developers to extend or substitute tools (Li et al., 2025c). This fragmentation increases the development burden for researchers seeking to experiment with novel tools or multi-tool scenarios. Second, these systems often rely on synchronous rollout mechanisms that process trajectories batch by batch (Jin et al., 2025). In such settings, the tool interaction phase is triggered only after all actions $a_i$ in a batch have been generated, resulting in idle bubbles and inefficient utilization of computational resources, especially on GPUs.

To address these issues, we propose VERL-TOOL, a general-purpose ARLT framework designed to support various tools as modular plugins via a unified API. Our goal is to minimize the integration overhead for community developers and provide a more efficient and extensible infrastructure for training LLMs with tool-use capabilities.

**Overview.** As shown in Figure 1, **VERL-TOOL** adopts a modular and decoupled architecture consisting of two main components: the **VeRL Workflow** and the **Tool Server**, connected via a unified API. This separation enables independent management of RL training and tool execution while preserving efficient communication between them.

The **VeRL Workflow** handles all reinforcement learning activities, including multi-turn rollouts and actor updates. The LLM actor interacts with the external environment by generating a sequence of actions $\{a_0, a_1, \dots\}$, each potentially triggering a tool interaction. Once an action is identified as tool-invoking (via matching a predefined stop token), it is sent to the **Tool Server** along with auxiliary metadata. The observation $o_i$ returned by the tool is then appended to the rollout, enabling observation-aware agent behavior and reward computation.

**Asynchronous Rollout Design.** A key feature of VERL-TOOL is its support for fully *asynchronous rollouts*, which avoids the inefficiency of traditional synchronous batch-based frameworks. In such a setting, tool calls are processed only after the entire batch has completed generating their respective actions $a_i$, resulting in idle "bubbles" in GPU and CPU utilization. In contrast, VERL-TOOL enables each trajectory to interact with the tool server independently and immediately upon finishing its action generation, as shown in Figure 2. This design ensures that tool execution latency does not block the entire batch, significantly improving throughput and system utilization in large-scale distributed settings. As shown in Table 3, the actor and environment evolve concurrently, achieving near 2 times speedup for the rollout stage.

```python
@register_tool
class BaseTool:
    tool_type = __name__
    def __init__(self, num_workers=4):
        self.num_workers = num_workers
        self.env_cache = {}

    def parse_action(self, action:str):
        # parse llm generated action and determine
            if match
        parsed_action = ...
        valid = True # invoke this tool if match (
            True),
        return parsed_action, valid

    def load_env(self, trajectory_id):
        env = ...
        return env

    def save_env(self, trajectory_id, env):
        ...

    def update_env(self, trajectory_id, env, **
        kwargs):
        ...

    def conduct_action(self, trajectory_id:str,
        action:str, extra_field:dict):
        parsed_action, is_valid = self.parse_action(
            action)
        # load current env
        env = self.load_env(trajectory_id)

        # get observation from parsed_action
        observation = ...

        done = True # if ending this trajectory

        # update and save current env
        self.update_env(trajectory_id, env,
            parsed_action, is_valid, extra_field,
            observation)
        self.save_env(trajectory_id, env)

        return observation, done, is_valid
```

```python
from .base import BaseTool, register_tool

def execute_python(code:str, timeout:int=60):
    # use subprocess to launch a python execution
    ...
    return stdout, stderr, has_error

@register_tool
class PythonCodeTool(BaseTool):
    tool_type = "python_code"

    def parse_action(self, action:str):
        # Try to find Python code in various formats
        all_valid_python_code = re.findall(r"<python
            >(.*?)</python>", action, re.DOTALL)

        if len(all_valid_python_code) == 0:
            return "", False
        # use all the code blocks
        parsed_code = "\n".join([code.strip() for
            code in all_valid_python_code])
        return parsed_code, True

    def conduct_action(self, trajectory_id:str,
        action:str, extra_field:dict):
        parsed_action, is_valid = self.parse_action(
            action)
        # load current env
        env = self.load_env(trajectory_id)

        # get observation from parsed_action
        stdout, stderr, has_error = execute_code(
            parsed_code)
        observation = stdout + "\n" + stderr

        done = False # do not end until llm reaches
            max turns or max length

        # update and save current env
        self.update_env(trajectory_id, env,
            parsed_action, is_valid, extra_field,
            observation)
        self.save_env(trajectory_id, env)

        return observation, done, is_valid
```

Figure 3: Example of code design for adding a new tool in VERL-TOOL via the plugin interface.

**Modular Tool-as-Plugin Design.** As illustrated in Figure 3, VERL-TOOL adopts a modular plugin system that cleanly abstracts tool interaction as an interface between the LLM actor and its external environment. Each tool is implemented as a subclass of a unified `BaseTool`, enabling seamless registration and extensibility. During rollouts, the actor's action $a_i$ is parsed by `parse_action` to determine whether it invokes a tool; valid calls are routed to the appropriate module, which retrieves the trajectory state via `load_env`. The tool then executes its `conduct_action`, returning the observation $o_i$, a validity flag, and a termination flag for next action generation. We also maintain per-trajectory environments through lightweight state dictionaries, updated via `update_env` and cleared at the end of an episode with `delete_env`. By decoupling tool logic from the training workflow, developers can add new tools with minimal overhead, while the framework dynamically manages their execution across threads or distributed workers.

**Tokenization.** A practical challenge in multi-turn agentic RL is how to tokenize tool observations and concatenate them with preceding LLM actions. Two strategies exist: (i) tokenize the action and observation strings separately, then append their token sequences; or (ii) concatenate the raw strings first and tokenize jointly. Although they yield the same sequence most of the time, discrepancies may arise for some specific combinations, as illustrated in Figure 4. The action string "`</python>`" and observation "`\n<result>`" produce consistent tokens under the first strategy (*On-Policy*), whereas the second one merges boundary symbols into a different token id (e.g., `29`,`198` vs. `397`), which changed the LLM-generated contents (*Off-Policy*). To avoid such inconsistencies, we adopt the first approach and always keep a consistent token list prefix during rollout, ensuring stable alignment in multiple rollouts turns.

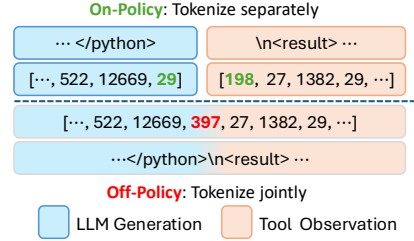

Figure 4: Tokenization of LLM generated content "...`</python>`" and tool observation "`\n<result>`..." can produce different token lists using Qwen2.5 tokenizer under different strategies.

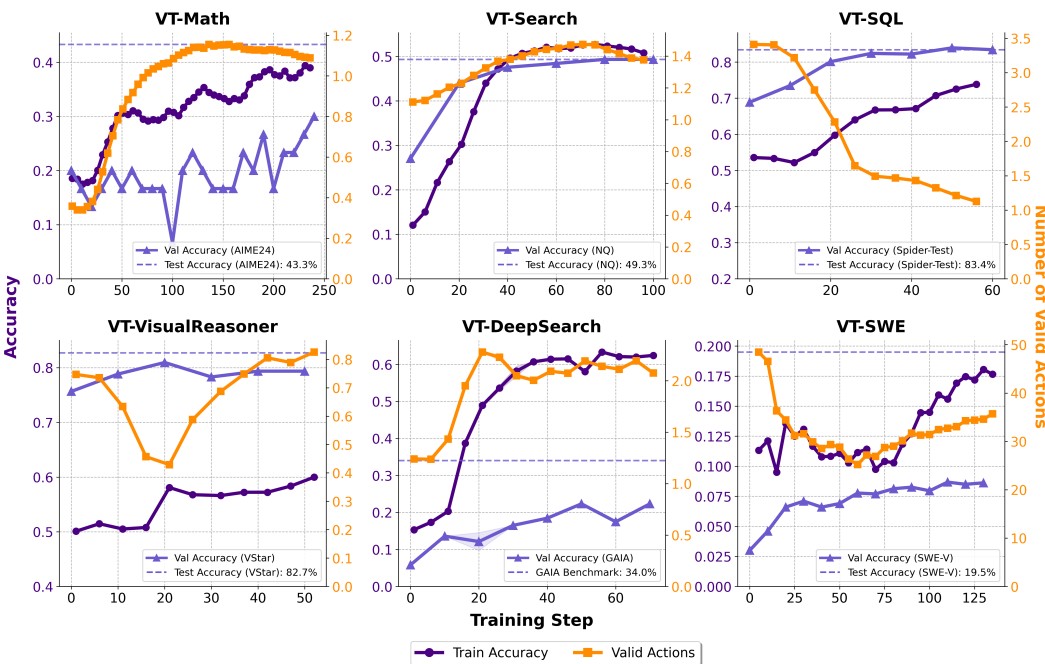

Figure 5: Training dynamics using VERL-TOOL on all 6 tasks. For each task, the corresponding test benchmarks are AIME24, NQ, Spider-Test, VStar, GAIA, and SWE-Verified. All models are trained and evaluated based on **VERL-TOOL** framework. The actual evaluation performance (purple dash) can be higher due to the train-eval settings difference. The number of actions is averaged over all sampled responses in each batch.

**Parallel Tool Server Backend.** To support high-throughput and scalable execution of tool interactions, VERL-TOOL offers two types of parallel execution backends within the Tool Server:

- *Multi-threading*: For small-scale or lightweight tool calls, VERL-TOOL employs Python's `ThreadPoolExecutor` to parallelize calls across multiple worker threads.
- *Ray-based Asynchronous Execution*: To deal with resource-intensive tools for better resource management, VERL-TOOL optionally supports integration with Ray (Moritz et al., 2017), enabling distributed and fault-tolerant tool execution across machines or GPU nodes. This design provides robust scalability for long-horizon or computationally intensive tools.

## 4 EXPERIMENTS

### 4.1 EXPERIMENT SETUP

With a modular, plug-and-play design, VERL-TOOL equips an agent with tools spanning multiple domains and modalities as shown in Table 8. In this section, we show the experiment results in six agentic RL with tool use (ARLT) tasks, including VT-Math (Table 4), VT-Search (Table 5), VT-SQL (Table 6), VT-VisualReasoner (Table 7), VT-DeepSearch (Table 7), and VT-SWE (Table 6), demonstrating the compatibility of VERL-TOOL with various tools. Please see details of training, evaluation in Appendix A.

### 4.2 RESULTS

**Training on VERL-TOOL achieves competitive results.** Models trained using VERL-TOOL consistently match or exceed existing baselines across all six tasks. VT-Math achieves a 62.2% average performance on mathematical benchmarks, surpassing expert models on multiple benchmarks such as AIME24, AMC23, and Olympiad Bench. VT-Search reaches 45.9% accuracy on knowledge QA, surpassing Search-R1 by 10.9%. In terms of the NL2SQL task, VT-SQL matches specialized sys-

Table 4: Results on Math-related benchmarks with Python interpreter tool. The best results are indicated in **bold** and the second-best results are underlined.

| Model | GSM8K | MATH 500 | Minerva Math | Olympiad Bench | AIME24 | AMC23 | Avg. |
|---|---|---|---|---|---|---|---|
| *Qwen2.5-Math-7B-Base/Instruct* | | | | | | | |
| Qwen2.5-Math-7B-Instruct | **95.2** | 83.0 | **37.1** | 41.6 | 16.7 | 70.0 | 57.3 |
| Qwen-2.5-Math-7B-Instruct-TIR | 88.8 | 80.2 | 26.8 | 41.6 | 30.0 | 52.5 | 53.3 |
| SimpleRL-Zoo-7B | 94.6 | 82.4 | 29.0 | 50.5 | 30.0 | 62.5 | 58.2 |
| ToRL-7B | 92.7 | 82.2 | 33.5 | 49.9 | 43.3 | 65.0 | 61.1 |
| VT-Math-zero (GRPO) | 91.8 | **83.2** | 31.6 | 50.5 | **43.3** | 70.0 | 61.7 |
| VT-Math-zero (DAPO) | 92.1 | 82.8 | 34.9 | **51.6** | 36.7 | **75.0** | 62.2 |

Table 5: Results of on knowledge-QA benchmarks. [†]/[⋆] represents in-domain/out-domain datasets.

| Model | General QA | | | Multi-Hop QA | | | | Avg. |
|---|---|---|---|---|---|---|---|---|
| | NQ[†] | TriviaQA[⋆] | PopQA[⋆] | HotpotQA[†] | 2wiki[⋆] | Musique[⋆] | Bamboogle[⋆] | Avg. |
| *Qwen2.5-7b-Base/Instruct* | | | | | | | | |
| Direct Inference | 13.4 | 40.8 | 14.0 | 18.3 | 25.0 | 3.1 | 12.0 | 18.1 |
| Search-R1-base (GRPO) | 39.5 | 56.0 | 38.8 | 32.6 | 29.7 | 12.5 | 36.0 | 35.0 |
| Search-R1-base (PPO) | 48.0 | 63.8 | 45.7 | 43.3 | 38.2 | **19.6** | 43.2 | 43.1 |
| VT-Search-zero (GRPO) | **49.3** | **66.2** | **50.2** | **44.8** | **45.3** | 19.3 | **46.4** | **45.9** |
| VT-Search-zero (DAPO) | 48.3 | 63.4 | 48.2 | 42.6 | 39.2 | 18.0 | 38.4 | 41.2 |

Table 6: Results on NL2SQL (left) and SWE-Verified (right) benchmarks in terms of pass rates.

| Model | Spider | |
|---|---|---|
| Split | Dev | Realistic |
| *Reasoning without Tool* | | |
| GPT-4o | 70.9 | - |
| DeepSeekCoder-6.7B-Instruct | 63.2 | - |
| OpenCoder-8B-Instruct | 59.5 | - |
| Qwen2.5-Coder-7B-Instruct | 73.4 | - |
| *Tool Integrated Reasoning* | | |
| OmniSQL-7B | 81.2 | 63.9 |
| SkyRL-SQL-7B (GRPO) | **83.9** | 81.1 |
| *VT-SQL (Qwen-2.5-Coder-7B-Instruct based)* | | |
| + GRPO | **83.9** | **81.3** |

| Model | SWEBench |
|---|---|
| *OpenHands Scaffold* | |
| Qwen3-8B | 3.6 |
| OpenHands-7B-Agent | 11.0 |
| *SkyRL-v0* (Cao et al., 2025) | |
| Qwen3-8B Based | 9.4 |
| OpenHands-7B-Agent Based | 14.6 |
| *R2E Gym Scaffold* | |
| Qwen3-8B | 10.4 |
| *VT-SWE (Qwen3-8B Based)* | |
| +GRPO | **19.5** |

Table 7: Results on visual reasoning (left) and agentic search benchmarks (right).

| Model | V* Bench |
|---|---|
| *Reasoning without Tool* | |
| GPT-4o | 62.8 |
| Gemini-2.5-Pro | 79.2 |
| Qwen2.5-VL-7B-Instruct | 70.4 |
| Video-R1-7B | 51.2 |
| *Tool Integrated Reasoning* | |
| Visual Sketchpad (GPT-4o) | 80.4 |
| IVM-Enhanced (GPT-4V) | 81.2 |
| Pixel-Reasoner-7B | **84.3** |
| *VT-VisualReasoner (Qwen2.5-VL-7B-Instruct Based)* | |
| + GRPO-Acc | 78.8 |
| + GRPO-Complex | 82.7 |

| Model | GAIA | HLE |
|---|---|---|
| *Reasoning without Tool* | | |
| DeepSeek-R1-671B | 25.2 | **8.6** |
| GPT-4o | 17.5 | 2.6 |
| Qwen3-8B | 20.4 | 4.6 |
| *Tool Integrated Reasoning (Qwen3-8B)* | | |
| Vanilla RAG | 20.4 | 5.8 |
| Search-o1 | 21.4 | 6.4 |
| WebThinker | 22.3 | 6.6 |
| ReAct | 23.3 | 4.6 |
| *VT-DeepSearch (Qwen3-8B Based)* | | |
| + Snippets-Only | 32.0 | 7.8 |
| + QwQ-32B | **34.0** | 8.4 |

tems such as SkyRL-SQL. VT-VisualReasoner achieves 82.7% on V* Bench while VT-DeepSearch reaches 34.0% on GAIA. These figures demonstrate that trained under our unified framework, agents could achieve competitive task-specific performance compared to separate divergent code bases.

**Multi-modal tools are well-supported.** VERL-TOOL's modular design enables the seamless integration of a wide range of types of tools within a unified API interface. We demonstrated this capability through the implementation of text-based tools support including Python and SQL Interpreter, Local Retriever and Web Search Tools, as well as visual processing tools that operate on image and video modality (image operations and video frame selections). Moreover, our framework is equipped with system-level tools such as the bash terminal and file operation tools. Visual reasoning experiments demonstrate that our agents could dynamically manipulate images and process visual information iteratively, enabling complex multi-modal workflows that were unsupported in existing single-modality RL-training frameworks, which only work on the text level.

**Dynamics of Tool Usage across Tasks.** Tool usage patterns exhibit substantial variation across different domains, with mathematical tasks typically requiring $1 \sim 4$ interactions while software engineering tasks may extend to over 100 interactions. Importantly, models do not spontaneously develop effective tool-use capabilities without appropriate reward design and initialization strategies. For instance, without VT-VisualReasoner's sophisticated reward mechanism, the frequency of tool actions gradually diminishes to zero within a few reinforcement learning (RL) steps.

The evolution of tool usage during training demonstrates task-specific characteristics that reflect the underlying utility of tool interactions. In VT-SQL settings, the number of actions decreases rapidly after several dozen RL steps, as the model learns that SQL executors are non-essential for most straightforward queries. Through training, the model develops a preference for responses requiring fewer tool calls by gradually memorizing expected execution results, thereby reducing the need for verification through SQL executors. Conversely, in VT-DeepSearch settings, tool usage increases dramatically during training because problem-solving fundamentally depends on search capabilities. Unlike simpler SQL results, the extensive and information-rich content returned by search tools cannot be easily memorized, necessitating increased tool invocations for effective task completion. This divergent behavior underscores how the intrinsic value of tool assistance shapes learning dynamics across different computational domains.

**Emerging behaviours of Agentic RL.** Agentic models trained using VERL-TOOL emerged sophisticated behaviors, including self-correction, iterative refinement, and strategic tool selection in the multi-round tool-calling environment provided by our framework. For instance, mathematical agents verify computations and backtrack from errors, search agents refine queries based on retrieved information, and software agents develop debugging strategies combining code analysis and incremental fixes. These capabilities represent genuine agentic problem-solving that extends beyond simple function-calling invocations. We present corresponding case studies in Appendix C.

## 5 CONCLUSION

We propose VERL-TOOL, addressing key limitations of Agentic Reinforcement Learning with Tool use (ARLT) models' training. Our framework features a unified and modular systematic design, providing multi-modal tool management through standardized API designs, while maintaining high-efficiency model training featuring asynchronous rollout execution. Our system extends traditional single-turn reinforcement learning with verifiable rewards to ARLT domains, featuring robust system designs and upstream-aligned with VeRL. The framework is extensively examined across six domains featuring diverse tool integrations and modalities. As evidenced by extensive evaluation, agents trained through our framework demonstrated competitive performance compared to specialized systems, while unified under our training infrastructure. We present VERL-TOOL as a scalable foundational training infrastructure to the RL community and hope our contributions could facilitate the advancement of ARLT research.

ETHICS STATEMENT

This study does not involve human subjects, sensitive data, or potentially harmful methodologies. The design and development of our framework adheres to principles of fairness, transparency, and reproducibility. We have reviewed ICLR 2026's Code of Ethics and confirm that our work complies with its guidelines in all aspects.

REPRODUCIBILITY STATEMENT

We affirm our commitment to the reproducibility of this research. The code is provided along with the paper submission, and our framework will be open-sourced and made publicly available upon publication to facilitate independent verification and further study. We put a full training setup in Appendix A and hyper-parameters in Table 9 to help better reproduce the experiment results.

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

APPENDIX

## A    DETAILED EXPERIMENT SETUP

We evaluate VERL-TOOL across six diverse domains to demonstrate its effectiveness in tool-augmented reasoning. Each task domain presents unique challenges and requires different tool integration strategies, allowing us to comprehensively assess the framework's adaptability and performance.

### A.1    MATHEMATICAL REASONING WITH PYTHON EXECUTOR (VT-MATH)

Mathematical reasoning tasks often involve complex computations that are prone to numerical errors when performed purely through natural language reasoning. To address this limitation, we integrate a Python code interpreter tool that enables agents to execute mathematical calculations reliably and verify intermediate results. We train a mathematical-coding agent that issues Python snippets to a sandboxed interpreter and processes execution traces.

We use DeepMath (He et al., 2025) as our training dataset. The reward function combines answer accuracy with tool usage incentives:

$$R_{\text{acc}}(\mathbf{x}, \mathbf{y}) = \begin{cases} 1 & \text{if match}(\mathbf{y}, \mathbf{y}_g) \\ -1 & \text{otherwise} \end{cases}, \quad R_{\text{tool}}(\mathbf{x}, \mathbf{y}) = \begin{cases} 0 & \text{if match}(\mathbf{y}, \mathbf{y}_g) \\ -0.25 & \text{otherwise} \end{cases} \quad (6)$$

where the final reward is $R_{\text{math}} = R_{\text{acc}}(\mathbf{x}, \mathbf{y}) + R_{\text{tool}}(\mathbf{x}, \mathbf{y})$. This design encourages the model to explore Python executor usage for problem-solving while maintaining an accuracy focus.

We evaluate on multiple mathematical benchmarks: MATH-500 (Hendrycks et al., 2021), OLYMPIAD (He et al., 2024), MINERVA (Lewkowycz et al., 2022), GSM8K (Cobbe et al., 2021), AMC, AIME24, and AIME25, using MATH-EVALUATION-HARNESS[1] for standardized eval.

### A.2    KNOWLEDGE QA WITH SEARCH RETRIEVER (VT-SEARCH)

Question answering tasks often require access to external knowledge beyond the model's parametric memory, particularly for factual queries and multi-hop reasoning. We integrate a FAISS-based search retriever tool that enables agents to query a local knowledge base and extract relevant information for answering complex questions.

Following prior work (Jin et al., 2025; Song et al., 2025), we integrate an E5 retriever (Wang et al., 2022) and index the 2018 Wikipedia dump (Karpukhin et al., 2020). The agent alternates between search operations and reasoning steps to construct comprehensive answers.

For this task, we apply accuracy as the primary reward:

$$R_{\text{search}}(\mathbf{x}, \mathbf{y}) = \begin{cases} 1 & \text{if match}(\mathbf{y}, \mathbf{y}_g) \\ -1 & \text{otherwise} \end{cases} \quad (7)$$

We evaluate using Exact Match scores on General Q&A benchmarks (NQ (Kwiatkowski et al., 2019), TriviaQA (Joshi et al., 2017), PopQA (Mallen et al., 2022)) and Multi-hop Q&A benchmarks (HotpotQA (Yang et al., 2018), 2Wiki (Ho et al., 2020), MuSiQue (Trivedi et al., 2022b), Bamboogle (Press et al., 2022)).

### A.3    MULTI-TURN SQL QUERY GENERATION (VT-SQL)

Natural language-to-SQL (NL2SQL) conversion requires understanding database schemas and translating natural language queries into executable SQL commands. This task benefits from tool integration as it allows iterative query refinement based on execution feedback and error correction.

We assess SQL Executor adaptability using the SkyRL-SQL training set (Liu et al., 2025) with Qwen2.5-7B-Instruct as the base model. The agent translates natural language questions into executable SQL, given schema hints and tool-calling instructions.

---

[1] https://github.com/ZubinGou/math-evaluation-harness

The reward function focuses solely on execution accuracy:

$$R_{\text{sql}}(\mathbf{x}, \mathbf{y}) = \begin{cases} 1 & \text{if match}(\mathbf{y}, \mathbf{y}_g) \\ -1 & \text{otherwise} \end{cases} \tag{8}$$

Following standard conventions, we evaluate execution accuracy (EX) on SPIDER-1.0 (Yu et al., 2018) (Dev and Test splits), SPIDER-DK (Gan et al., 2021b), and SPIDER-SYN (Gan et al., 2021a).

### A.4 VISUAL REASONING WITH IMAGE OPERATIONS (VT-VISUALREASONER)

Traditional visual reasoning tasks are conducted primarily in the text modality, where models cannot dynamically process images as actions. To address this limitation, we implement image operation tools that enable agents to zoom into specific image regions, select key frames, and perform other visual manipulations to enhance reasoning over dense visual information, following the Pixel-Reasoner (Su et al., 2025) approach.

The reward design incorporates both accuracy-oriented and compositional complexity measures:

$$R_{\text{visualreasoner}}(\mathbf{x}, \mathbf{y}) = r(\mathbf{x}, \mathbf{y}) + \alpha \cdot r_{\text{curiosity}}(\mathbf{x}, \mathbf{y}) + \beta \cdot r_{\text{penalty}}(\mathbf{y}), \tag{9}$$

$$\text{where} \quad r_{\text{curiosity}}(\mathbf{x}, \mathbf{y}) = \max(H - \text{RaPR}(\mathbf{x}), 0) \cdot \mathbf{1}_{\text{PR}}(\mathbf{y}) \tag{10}$$

$$r_{\text{penalty}}(\mathbf{y}) = \min(N - \mathbf{n}_{\text{vo}}(\mathbf{y}), 0) \tag{11}$$

where $\text{RaPR}(\mathbf{x})$ denotes the ratio of responses that invoke tool calls and $\mathbf{n}_{\text{vo}}(\mathbf{y})$ denotes the number of actions that response $\mathbf{y}$ invokes. Hyperparameters are set as $H = 0.3$, $N = 1$, $\alpha = 0.5$ and $\beta = 0.05$. We train two variants using accuracy-only reward and the original complexity-driven reward, denoted as "GRPO-acc" and "GRPO-complex" respectively.

We use the official training dataset from Pixel-Reasoner and evaluate primarily on V-Star (Wu & Xie, 2024), which assesses MLLM visual search capabilities.

### A.5 AGENTIC WEB SEARCH (VT-DEEEPSEARCH)

Open-web question answering requires real-time information retrieval and multi-step reasoning over diverse web sources. GAIA (Mialon et al., 2023) and HLE (Phan et al., 2025) are representative benchmarks testing these capabilities. We implement a Web Search tool using Google Search API through SERPER with caching, enabling agents to perform dynamic information gathering and synthesis from online sources.

We apply both accuracy and tool-usage rewards to encourage effective search behavior:

$$R_{\text{deepsearch}}(\mathbf{x}, \mathbf{y}) = R_{\text{acc}}(\mathbf{x}, \mathbf{y}) + R_{\text{tool}}(\mathbf{x}, \mathbf{y}), \quad \text{where } R_{\text{tool}}(\mathbf{x}, \mathbf{y}) = \begin{cases} 0.1, & \text{if tool is called} \\ 0, & \text{if no tool call} \end{cases} \tag{12}$$

We use 1K mixed training examples from SimpleDeepSearcher (Sun et al., 2025) and Web-Sailor (Li et al., 2025b), following the setting in Dong et al. (2025). Starting from Qwen3-8B, we evaluate on GAIA and HLE (text-only) benchmarks. We retrieve top-k URLs for each query and use the returned snippets as content during RL training. For evaluation, we employ two settings: "Snippet-Only" aligns with training conditions using only snippet content, while "QwQ-32B" uses a browser agent to summarize raw content from retrieved URLs.

### A.6 SOFTWARE ENGINEERING BENCHMARK (VT-SWE)

Software engineering tasks require code understanding, localization, debugging, and modification capabilities that benefit from iterative execution and testing. We integrate bash terminal and code execution tools to enable agents to interact with software development environments effectively.

We build on the R2E-Gym scaffold (Jain et al., 2025) and its training dataset R2E-Lite, using Qwen3-8B in no-think mode as the base model. The reward function is defined strictly by task completion accuracy: an agent must terminate normally and pass all verification tests to receive a reward of 1; otherwise, the reward is 0:

$$R_{\text{swe}}(\mathbf{x}, \mathbf{y}) = \begin{cases} 1 & \text{if execution terminates successfully and all tests pass} \\ 0 & \text{otherwise} \end{cases} \tag{13}$$

We evaluate performance on the SWE-Verified benchmark, measuring the agent's ability to resolve software engineering tasks and pass verification tests.

For training and evaluation, we maintain a cluster of eight servers (each with 64 CPU cores and 200 GB memory), orchestrating sandbox services via Kubernetes. Each task runs inside the official Docker image provided by R2E-Lite, initialized with 1 CPU and 2 GB memory, elastically scalable up to 2 CPUs and 4 GB memory. We observed that the main bottleneck lies in disk I/O during Docker initialization. To stabilize training, we therefore allocate more CPU and memory resources than are minimally required.

The modular architecture of VERL-TOOL, which separates training from environment services, allows us to scale sandbox environments efficiently. Each environment interaction is given a 90-second timeout, reward computation has a 300-second timeout, and the maximum time per trajectory is capped at 20 minutes. Any trajectory that times out, encounters an exception, or exceeds the length limit is assigned a reward of 0, and its gradients are masked during updates.

## A.7 SUPPORTED TOOLS

Table 8: Currently supported tools of VERL-TOOL framework.

| Tools | Description | Related works |
|---|---|---|
| 🐍 Code Interpreter | Execute Python code | ToRL (Li et al., 2025c) |
| 🔍 Faiss Search | Vector similarity search for documents | Search-R1 (Jin et al., 2025) |
| 🌐 Web Search API | Real-time web search and retrieval | SimpleDeepSearch (Sun et al., 2025) |
| 🖼 Image Processing | Image resize, video frame selection | PixelReasoner (Su et al., 2025) |
| >_ Bash Terminal | Execute shell commands | R2E-Gym (Jain et al., 2025) |
| 🗄 SQL Executor | Database queries and data management | SkySQL (Liu et al., 2025) |
| 🔌 MCP Interface | Model Context Protocol for external tool | ToolRL (Feng et al., 2025) |

## A.8 TRAINING AND EVALUATION CONFIGURATIONS

Table 9 summarizes the detailed configurations for each task during training and evaluation. Due to configuration differences across tasks, there may be gaps between validation curves and final downstream evaluation performance, as illustrated in Figure 5.

The RL training parameters vary across tasks to accommodate different complexity levels and interaction patterns. Math-TIR and Pixel-Reasoner use smaller batch sizes due to computational constraints, while Search-R1 employs larger batch sizes for stable retrieval learning. The agentic tool use parameters reflect task-specific requirements: Math-TIR typically requires single-turn interactions, while SWE tasks may require up to 100 interaction turns for complex debugging scenarios.

Evaluation parameters are configured to balance comprehensive assessment with computational efficiency. Temperature settings range from 0.0 for deterministic tasks like SQL generation to 0.6 for creative tasks requiring exploration. Maximum turn limits reflect task complexity, with simple QA tasks limited to 2-5 turns while software engineering tasks allow up to 100 turns for thorough problem resolution.

Table 9: Training and evaluation configurations across all six tasks.

| Tasks | VT-Math | VT-Search | VT-SQL | VT-VisualReasoner | VT-DeepSearch | VT-SWE |
|---|---|---|---|---|---|---|
| **RL Training Parameters** | | | | | | |
| Rollout BS | 128 | 512 | 256 | 128 | 128 | 32 |
| N Samples | 16 | 16 | 5 | 8 | 16 | 8 |
| Gradient BS | 128 | 64 | 256 | 128 | 128 | 32 |
| Temperature | 1.0 | 1.0 | 0.6 | 1.0 | 1.0 | 1.0 |
| Top P | 1.0 | 1.0 | 0.95 | 1.0 | 1.0 | 1.0 |
| Learning Rate | 1e-6 | 1e-6 | 1e-6 | 1e-6 | 1e-6 | 2e-6 |
| Val Temperature | 0.0 | 0.0 | 0.0 | 0.0 | 0.0 | 0.0 |
| Val Top P | 1.0 | 1.0 | 0.95 | 1.0 | 1.0 | 1.0 |
| **Agentic Tool Use Parameters** | | | | | | |
| Max Turns | 1 | 2 | 5 | 3 | 5 | 100 |
| MTRL | ✗ | ✗ | ✗ | ✔ | ✗ | ✔ |
| Max Prompt Length | 1024 | 4096 | 4096 | 16384 | 2048 | 10240 |
| Max Response Length | 3072 | 4096 | 4096 | 16384 | 8196 | 22528 |
| Max Action Length | 2048 | 2048 | 2048 | 2048 | 8196 | 10240 |
| Max Observation Length | 512 | 1024 | 1024 | 8192 | 4096 | 10240 |
| Action Stop Tokens | ```output | </search>, </answer> | </sql> | </tool_call> | </python>, </search> | </function> |
| **Evaluation Parameters** | | | | | | |
| Temperature | 0.6 | 0.0 | 0.0 | 0.0 | 0.6 | 1.0 |
| Top P | 0.95 | 1.0 | 1.0 | 1.0 | 0.95 | 1.0 |
| Max Turns | 4 | 2 | 5 | 5 | 10 | 100 |
| Max Prompt Length | 1024 | 4096 | 4096 | 16384 | 2048 | - |
| Max Response Length | 3072 | 4096 | 4096 | 16384 | 32768 | 40960 |
| Max Action Length | 2048 | 2048 | 2048 | 4096 | 16483 | - |
| Max Observation Length | 512 | 1024 | 1024 | 8192 | 4096 | - |

## B    MORE RELATED WORKS

### B.1    ORGANIZATION

In this section, we first establish the definition of Agentic Large Language Models (Agentic LLMs). Then, prior tool integration efforts in LLMs and the shift from single-turn, prompt-driven tool-calling to instruction tuning/RL-based multi-turn agentic interaction are reviewed. Further, we introduce various representative domain-specific tasks that are proven to benefit from developing corresponding tool-use oriented agents. We then distinguish Reinforcement Learning with Verifiable Rewards (RLVR) from Agentic Reinforcement Learning with Tool Use (ARLT). Finally, we survey existing systems for training RL-based tool-using agents and position our work: VERL-TOOL.

### B.2    TOOL-INTEGRATED REASONING

Augmenting Large Language Models (LLMs) with external tools has become a prominent approach to address limitations in parametric reasoning and enable more complex task solving (Shen, 2024). Early approaches focused on *prompt-based integration*, where systems like HUGGINGGPT (Shen et al., 2023), CHAMELEON (Lu et al., 2023), and MULTITOOL-COT (Inaba et al., 2023) used tool manuals, demonstrations, or structured Chain-of-Thought templates to orchestrate sequential tool invocations. While these methods offer plug-and-play convenience, they face challenges when adapting to complex, multi-step tasks due to their reliance on static prompting strategies.

A complementary line of work explores *instruction tuning*, where models are explicitly trained to recognize tool formats and generate appropriate function calls. Representative systems include TOOLFORMER (Schick et al., 2023), which uses bootstrapped annotations to teach tool usage patterns, GPT4TOOLS (Yang et al., 2023), which distills tool-use traces from more capable models, and LIMO (Ye et al., 2025), which demonstrates that targeted examples can elicit extended reasoning chains. However, these supervised approaches primarily provide static guidance and lack mechanisms for dynamic error correction based on tool execution feedback.

In contrast, reinforcement learning approaches enable models to develop adaptive tool-calling strategies through interaction-based training. Our work builds on this direction by employing GRPO training to enhance models' capabilities for self-reflection and iterative refinement in response to tool feedback.

### B.3    FROM TOOL INTEGRATION TO AGENTIC LLMS

Large language models (LLMs) demonstrated exceptional flexibility and generality with model parameter and training-data scaling (Team et al., 2024; Qwen et al., 2025; OpenAI, 2025). With recent research advancements dramatically enhancing their capability in reasoning, information retrieval, and instruction-following (Ke et al., 2025), current research trends have shifted from developing simple LLM-based tool-callers to empowering LLMs with versatile interaction capability to facilitate their agentic actions in the world (Plaat et al., 2025).

**Agentic LLMs.**    As one of the central concepts in artificial intelligence (Russell & Norvig, 2016), Decision-making, identifying environmental changes, communication, and acting on one's goal or will are generally defined as common traits of Agency (Epstein & Axtell, 1996; Wooldridge, 1999; Gilbert, 2019). Following well-established conventions, we denote Agentic LLMs as models that reason, act through tools, and interact over one or multiple turns, maintaining state and revising plans according to environmental observations, framing agent behavior beyond passive text-generation, and motivating the possession of improved planning and tool-use capabilities.

**Agentic Tool-calling Acquisition.**    As one fundamental capability, tool-calling has been widely studied as one of the most effective ways of empowering Agentic LLMs with the capability to act in exposed environments. Early attempts involve prompt-based orchestration, which structures function calls in a training-free manner with specific instructions and tool-calling schemas (Yao et al., 2022; Lu et al., 2023), exploiting LLMs' instruction-following capability through Chain-of-Thought technique (Wei et al., 2022) and explicit task decomposition (Kim et al., 2023; Huang et al., 2022) or multi-agent orchestration (Shen et al., 2023; Ruan et al., 2023). Instruction-tuning

based tool callers learn function-call schemas and appropriate tool-call choices through supervised traces (Schick et al., 2023; Kong et al., 2023; Gou et al., 2023), improving problem-solving capabilities while suffering from a lack of generality and remaining largely single-turn (Qu et al., 2024). Reinforcement-Learning-based acquisition further enhances tool-using generality and multi-turn behaviors (Li et al., 2025c; Feng et al., 2025) through problem-solving outcomes and tool-feedback, facilitating environment exploration, self-reflection, error-corrections, and enhanced performance on reasoning-intensive tasks through informative tool responses. (Moshkov et al., 2025).

### B.4 REINFORCEMENT LEARNING WITH VERIFIABLE REWARD (RLVR)

Conventional RLHF pipelines, such as DPO (Rafailov et al., 2023) and PPO (Schulman et al., 2017), optimize answer-level quality and require reward models with substantial sizes. With the introduction of GRPO in DEEPSEEKMATH (Shao et al., 2024), as a critic-free variant of PPO, it enables stabilized RL training on long reasoning chains and enables the integration of multi-turn tool responses and verifiable rewards in reinforcement learning. Existing works have explored the potential of extending GRPO training through the inclusion of rule-based verifiable rewards such as format-based rewards and exact-match based comparisons, as well as tool-calling responses. They demonstrated significant success in developing expert agents in multiple domains with tool-calling and self-reflection capabilities.

By integrating GRPO and verifiable rewards, advancements have been witnessed in a wide range of domain-specific tasks. TORL (Li et al., 2025c) integrates Python Code Interpreters into the GRPO training of solving mathematical tasks, surpassing RL baselines which does not have tool-calling integration on multiple mathematical datasets. TOOLRL (Qian et al., 2025) explored and analyzed the impact of reward design and tool choice on the effect of tool-integrated GRPO training, revealed the significance of tool-use reward design in boosting LLMs' tool calling and generalization capability, as well as achieving stable GRPO training. RETOOL (Feng et al., 2025) integrated tool-calling in PPO training, resulting in a Python tool-enabled agent with strong mathematical problem-solving capability. EXCOT (Zhai et al., 2025) and THINK2SQL (Papicchio et al., 2025) demonstrated preliminary success of utilizing GRPO and the response of SQL executors to enhance the base model's performance on natural language-to-SQL (NL2SQL) tasks through comprehensive reward design and challenging training problem-filtering. By integrating Faiss (Douze et al., 2024)-based local retrievers and API-based online search services, SEARCH-R1 (Jin et al., 2025) and R1-SEARCHER (Song et al., 2025) equip base models with enhanced search tool-calling and retrieval capability, achieving superior performance across multiple retrieval-centric benchmarks.

### B.5 AGENTIC REINFORCEMENT LEARNING WITH TOOL USE (ARLT)

Despite RLVR optimizing policies' behaviors using rule-based verifiable checks with the inclusion of single-turn tool-calling, current RLVR-based agentic models' limited capability in making real multi-turn, long-horizon interactions with tools and perform effective self-reflection and dynamic plan revising remains to be the gap to the generalist Agent-LLM. The inclusion of training the model over dynamic, long-term interactions while exposing it to intermediate tool responses could shape the agentic model's subsequent actions, pushing beyond single-turn verification toward long-horizon, interaction-centric RL training.

Therefore, we use Agentic Reinforcement Learning with Tool Use (ARLT) to denote reinforcement learning on dynamic, multi-turn trajectories in which tool responses are treated as environmental observations that condition future actions. ARLT therefore (i) assigns credit across tool calls rather than only at the final answer, (ii) handles observation tokens explicitly (e.g., masking non-model tokens during GRPO optimization), and (iii) relies on asynchronous, failure-aware executors for adapting potentially slow, stochastic, or error-prone (Plaat et al., 2025; Ke et al., 2025) tool calls. In contrast to RLVR, ARLT targets exploration, re-planning, and recovery from tool failures, and is naturally suited to settings where the problem-solving requires probing the environment before coming up with correct solutions. In this work, our framework is mainly evaluated on the following domain-specific tasks.

**Mathematical Interactive Coding.** Tool-integrated reasoning was first introduced to tackle computationally intensive mathematical problems by combining natural language reasoning with pro-

gramming strategies (Chen et al., 2022; Yue et al., 2023; Jin et al., 2025; Song et al., 2025; Wang et al., 2024a; Chen et al., 2022). Building on this idea, Wang et al. (2023) proposed an iterative method that couples textual reasoning with code execution to cross-check the answers, improving the accuracy. More recently, Chen et al. (2025) incorporated code execution into reasoning through supervised fine-tuning on curated code-integrated CoT data. Yet this method is limited by its dependence on specific data distributions and cannot learn adaptive tool-use strategies—such as when and how to call tools—via reinforcement learning. To solve this, concurrent work, including ToRL (Li et al., 2025c) and ZeroTIR (Mai et al., 2025), applies ZeroRL to train agents for mathematical code interpreter use.

**Agentic Search and Retrieval.** Large language models (Gemini, 2024; OpenAI, 2025) possess a huge amount of intrinsic knowledge while struggling at domain-specific, knowledge-centric tasks (Chen et al., 2024b; Peng et al., 2023) and suffer from hallucination (Zhang et al., 2023; Huang et al., 2023). A common approach to mitigate this issue is by integrating search engines into the LLMs. Predominant approaches of search engine integration often fall within two categories: Retrieval Augmented Generation (RAG)-based (Lewis et al., 2020; Gao et al., 2023b) and Tool-calling based retrieval (Schick et al., 2023). As RAG relies on a separate retriever to extract documents in a single turn without the interaction of LLM, it faces challenges of retrieving irrelevant information or returning less useful context (Jin et al., 2024; Jiang et al., 2023). Conversely, Tool-calling-based retrieval enhances LLMs' capability of calling the search retriever as a tool either through prompting (Yao et al., 2022; Trivedi et al., 2022a), fine-tuning (Schick et al., 2023), or through reinforcement learning (Jin et al., 2025; Song et al., 2025; Jiang et al., 2025), while enhancing searching agents with multi-turn tool-calling-based retrieval capability remains under-explored.

**Natural Language to SQL.** Natural language-to-SQL (NL2SQL) refers to the task of generating database-specific query codes for extracting data of interest. Early efforts typically involve developing expert models with an encoder-decoder structure for achieving this goal, where the encoder fuses the database schema and the natural-language query, leaving the answer generation to the decoder module (Wang et al., 2019; Cai et al., 2021; Li et al., 2023; Liu et al., 2024). With the advances of sequence-to-sequence models, especially large language models, the NL2SQL tasks have seen a rapid transfer from simple sequence-to-sequence modeling into developing tabular understanding agents utilizing pre-trained large language models. Existing works mostly focus on prompt enhancement (Gao et al., 2023a), large-scale data synthesis and model finetuning (Li et al., 2024; 2025a), while some recent works explored the realm of Reinforcement Learning-driven NL2SQL agents (Papicchio et al., 2025; Pourreza et al., 2025) and NL2SQL agents featuring multi-round Tool Calling capabilities. (Liu et al., 2025)

## C  CASE STUDY

In this section, we present case studies of representative tasks, featuring corresponding agents' outputs to illustrate Multi-Turn Tool Calling. The queries are randomly extracted from the underlying benchmarks. For readability, tag markers are colorized according to their functional categories:

1. Internal reasoning tags, `<think>...</think>` are colored in purple.

2. Tool-calling tags are formatted as `<python>/<sql>/<search>`.

3. Observation/Tool Response tags are colored as `<observation>/<information>/<result>`.

4. The final output tags are formulated as `<answer>/<solution>`.

We specifically chose the following four representative tasks for demonstration: **Mathematical Reasoning with Python Executor**, **Knowledge Q&A with Search Retriever**, **Multi-turn SQL Generation**, and **Agentic Web Search**. For a detailed explanation of tag definitions, please refer to Tab.9 and corresponding settings in Appendix's Sec. A.

For Agentic Web Search, Multi-turn SQL Generation, and Knowledge Q&A with Retriever, the text-heavy responses from the tools are truncated or simplified.

Table 10: Example answer demonstration of VT-MATH

**Example from DeepMath Dataset**

**Question:**
Determine the function $f$ given that it has a power series representation centered at 0, converging within the interval $(-1, 1)$, and there exists a sequence of distinct points $(x_n)$ approaching 0 such that $f(x_n) = 0$ for all $n \in \mathbb{N}$. What is $f$?

**Labeled Answer:**
0

**Model Output by VT-Math GRPO**

Let's consider the problem. The function $f$ has a power series representation centered at 0 that converges within the interval $(-1, 1)$. This means that $f(x)$ can be written as: $f(x) = \sum_{n=0}^{\infty} a_n x^n$ where the series converges for $|x| < 1$. We are given that there exists a sequence of distinct points $(x_n)$ approaching 0 such that $f(x_n) = 0$ for all $n \in \mathbb{N}$. This implies that: $f(x_n) = \sum_{n=0}^{\infty} a_n x_n^n = 0$ Since the series converges at $x_n$ and $x_n \to 0$, we can conclude that each $a_n = 0$ for all $n \geq 1$ because if any $a_k \neq 0$ for some $k \geq 1$, then the series would not converge to 0 as $x_n \to 0$. Therefore, the only possible term is the constant term $a_0$. This means we can write:

$$f(x) = a_0$$

Given that $f(0) = a_0 = 0$ (since $f(x_n) \to f(0) = 0$ as $x_n \to 0$), we have:

$$f(x) = 0$$

Therefore, the function $f$ is the zero function. Let's put this understanding into a more formal version:

```python
from sympy import symbols, Function, Eq, solve

# Define the variable and the function
x = symbols('x')
f = Function('f')(x)
# Define the power series representation centered at 0
# Since we are given that there exists a sequence (x_n) approaching 0
    such that f(x_n) = 0 for all
# It implies that the function is the zero function
f_zero = 0
# Print the zero function
print(f_zero)
```
```output
result
0
```

The power series representation of the function $f$ converging within the interval $(-1, 1)$, given that there exists a sequence of distinct points $(x_n)$ approaching 0 such that $f(x_n) = 0$ for all $n \in \mathbb{N}$, results in the function being the zero function. Therefore, the function $f$ is $\boxed{0}$.

Table 11: Example answer demonstration of VT-SEARCH

---

**Example from HotpotQA Dataset**

---

**Question:**
What is the birth name of Nadeem Siddique's favorite boxer?
**Labeled Answer:**
Walker Smith Jr.

---

**Model Output by VT-Search (GRPO)**

<think>
To answer this question, I need to find out Nadeem Siddique's favorite boxer and then determine the birth name of that boxer.
</think>
<search>
Who is Nadeem Siddique's favorite boxer?
</search>
<information>
Doc 1(Title: "Nadeem Siddique") Nadeem Siddique Siddique was born in Bradford, Yorkshire, United Kingdom. He supports his local football team Bradford City and **his favorite boxer is Sugar Ray Robinson**...
Doc 2(Title: "Nadeem Siddique") in 2005 for the people of Palestine and Gaza whereby he raised £352,000. He stated 'Me and Junior have been doing loads of things for charities and with our popularity in Bradford we always raise quite a few quid...
Doc 3(Title: "Nadeem Siddique") 3 fights and 3 wins and is on the verge of a major championship fight. Siddique is the founder of 'Make the Weight' gym; the opening of the gym was attended by boxer Mike Tyson in whose...
</information>
From the search results, I found that Nadeem Siddique's favorite boxer is Sugar Ray Robinson.
<search>
What is the birth name of Sugar Ray Robinson?
</search>
<information>
Doc 1(Title: "Sugar Ray Robinson") Robinson was born **Walker Smith Jr.** in Ailey, Georgia, to Walker Smith Sr. and Leila Hurst. Robinson was the youngest of three children...
Doc 2(Title: "Sugar Ray Robinson") disease. He died in Los Angeles at the age of 67 and was interred in the Inglewood Park Cemetery, Inglewood, California. Robinson married Marjorie Joseph in 1938; the marriage was annulled the same year. Their son, Ronnie Smith...
Doc 3(Title: "Sugar Ray Robinson") Jackie Gleason, Nat King Cole, Joe Louis, and Lena Horne among others. Robinson was known as a flamboyant personality outside the ring. He combined striking good looks with charisma and a flair for the dramatic...
</information>
From the search results, I found that the birth name of Sugar Ray Robinson is Walker Smith Jr.
<answer> Walker Smith Jr. </answer>

Table 12: Example answer demonstration (SQL reasoning)

**Example (Student–Pet DB)**

**Question:**
What are the ids of the students who do not own cats as pets?
**Labeled Answer:**

```sql
SELECT stuid FROM student EXCEPT SELECT T1.stuid FROM student AS T1 JOIN
    has_pet AS T2 ON T1.stuid  =  T2.stuid JOIN pets AS T3 ON T3.petid
    =  T2.petid WHERE T3.pettype  =  'cat'
```

**Model Output (Agentic SQL)**

**\<think\>** I need to find the student IDs of students who do not own cats as pets. To achieve this, I will first identify all students who own cats and then exclude them from the list of all students.
**\</think\>**
**\<sql\>**

```sql
SELECT StuID FROM Has_Pet WHERE PetID IN (SELECT PetID FROM Pets WHERE
    PetType = 'cat');
```

**\</sql\>**
**\<observation\>**
0
1001
\<reminder\>You have 5 turns left to complete the task.\</reminder\>
**\</observation\>**
**\<think\>** I have the list of student IDs who own cats. Now, I will use a NOT IN clause to exclude these IDs from the list of all students.
**\</think\>**
**\<sql\>**

```sql
SELECT StuID FROM Student WHERE StuID NOT IN (SELECT StuID FROM Has_Pet
    WHERE PetID IN (SELECT PetID FROM Pets WHERE PetType = 'cat'));
```

**\</sql\>**
**\<observation\>**
0
1002
1003
...
\<reminder\>You have 4 turns left to complete the task.\</reminder\>
**\</observation\>**
**\<think\>** I have the list of student IDs who do not own cats as pets. Since the question asks for the ids of the students, I can now output the final SQL as the solution.
**\</think\>**
**\<solution\>**

```sql
SELECT StuID FROM Student WHERE StuID NOT IN (SELECT StuID FROM Has_Pet
    WHERE PetID IN (SELECT PetID FROM Pets WHERE PetType = 'cat'));
```

**\</solution\>**

Table 13: Example answer demonstration of VT-DEEPSEARCH

**Example from Gaia Dataset**

**Question:**
Find the value of $x$ to the nearest tenth: Lx = (d/dx * (A * x-squared)) + 4-thousand'n'ninety-7 minus C
Where L is the last two digits of the year of the Venezuelan Declaration of Independence, A is the number of colors in the TikTok logo as of July 2023, excluding black and white, and C is the height of the average woman in the Philippines according to a July 2023 Business Insider article, rounded to the nearest whole centimeter.

**Labeled Answer:**
563.9

**Model Output by VT-DeepSearcher (GRPO)**

**<think>** Okay, let's try to figure out this problem step by step. First, I need to understand what each part of the equation means and find the values for L, A, and C. Then I can plug them into the equation to solve for x.

The equation given is Lx = (d/dx * (A * x-squared)) + 4-thousand'n'ninety-7 minus C. Wait, the wording here is a bit confusing. Let me parse it again. Maybe it's L multiplied by x equals the derivative of (A times x squared) plus 4097 minus C. Or maybe there's a typo? Let me check the original question again.

The equation is written as: Lx = (d/dx * (A * x-squared)) + 4-thousand'n'ninety-7 minus C. Hmm, the "4-thousand'n'ninety-7" part might be a way to write 4097. So maybe it's 4097. So the equation is Lx equals the derivative of A x squared plus 4097 minus C.

**<search>** Venezuelan Declaration of Independence year **</search>**
**<result>**
The Venezuelan Declaration of Independence occurred on **July 5, 1811**. The last two digits of this year are **11**, which corresponds to L...
**</result>**
So L = 11.

**<search>** TikTok logo colors 2023 **</search>**
**<result>** The TikTok logo as of 2023 features **two colors excluding black and white** : pink (#FE2C55) and aqua/cyan (#25F4EE). Thus A = 2...
**</result>**
So A = 2.

**<search>** average height of women in Philippines 2023 Business Insider **</search>**
**<result>** The average height of women in the Philippines is **149.6 cm**, rounded to **150 cm**. Thus C = 150...
**</result>**
So C = 150.
Now the equation is:
$Lx = (d/dx(Ax^2)) + 4097 - C$.
Derivative of $Ax^2$ is $2Ax = 4x$.
Substituting $L = 11, A = 2, C = 150$:
$11x = 4x + 4097 - 150 = 4x + 3947$.
Simplify: $7x = 3947 \Rightarrow x = 3947/7$.
**<python>**

```
3947 / 7
```

**</python>**
**<result>**
563.9
**</result>**
**<answer>**
The final answer is 563.9 .
**</answer>**
**</think>**

## D  THE USE OF LARGE LANGUAGE MODELS

We have utilized Claude and ChatGPT's website clients primarily for writing enhancement tasks, including eliminating potential grammar errors, improving the logical coherence of existing content, and reorganizing tables and figures for better presentation. Additionally, we employed these tools for data visualization, such as generating Figure 5 from our collected data. We also leveraged ChatGPT's deep research capabilities to identify relevant works in agentic RL and infrastructure, which are comprehensively presented in section 2 and Appendix B.

