# Rebuttal Supplementary Materials for ICLR 2026 submission:
# VerlTool: Towards Holistic Agentic Reinforcement Learning with Tool Use

# 1 GENERAL RESPONSE TO ALL REVIEWERS AND AC

We thank all reviewers for their thoughtful feedback and are pleased that they recognized several strengths of our work:

- **Well-designed and efficient framework with unified APIs:** Reviewers highlighted that our system provides "much-needed orchestration and glue code" for tool-augmented roll-outs and is "well-designed and, if well-maintained," will have long-term impact.

- **Broad tool coverage and comprehensive evaluation:** Our framework supports a wide range of *stateful, multimodal* tools and demonstrates strong empirical performance across six real agentic RL tasks.

- **Reproducibility and practical insights:** Reviewers appreciated our extensive documentation, practical considerations (e.g., tokenization), and clear discussion on reproducibility.

- **Clear writing and strong presentation:** The paper received an average presentation score of **3.5 (Top 1.59%)**.

The paper also ranks strongly in overall metrics—soundness **2.75 (Top 20.58%)** and contribution **3.25 (Top 2.52%)**—indicating substantial and broadly useful contributions. Statistics come from Paper Copilot statistics.

To address all concerns comprehensively, we provide a consolidated summary of new experiments, analyses, and clarifications added to this rebuttal:

- **Clarification of positioning vs. other RL framework like Verl-MCP, AREAL, and SkyRL:** We articulate fundamental architectural differences in statefulness, multimodal support, and resource management, clarifying why prior systems cannot be directly extended.
  *(Requested by UHTW, YLgB, PUT8, see in section 2.)*

- **Cross-framework efficiency comparison:** We benchmark per-step wall-clock time across frameworks and show Verl-Tool's advantages.
  *(Requested by UHTW, see in subsection 2.3.)*

- **Details of VERLTOOL's error handling strategies:** We expand more details on error handling (timeouts, retrying, penalization, masking), trajectory-level recovery, and tool-server stability.
  *(Requested by UHTW, PUT8, see in section 3.)*

- **Tool-Server efficiency benchmarking and stability analysis:** We include concurrency benchmarks showing stable memory, latency, and throughput under heavy loads.
  *(Requested by PUT8 and PUT8, see in section 4.)*

- **Ablation studies of key design choices:** We include new ablations on tokenization stability and tool-server efficiency—showing that avoiding retokenization significantly improves training stability.
  *(Requested by UHTW, zwhJ, PUT8, see in section 5.)*

- **Multi-tool RL experiments:** We add new experiments combining Math–SQL and Math–Visual tool use within the same trajectory and analyze when multi-tool setups succeed or collapse.
  *(Requested by zwhJ, YLgB, see in section 6.)*

- **Clarification of VT-\* improvements over baselines:** We analyze differences in training parameters, tool-calling formats, and maximum response lengths to explain occasional improvements over official baselines.
  *(Requested by UHTW, see in section 7.)*

- **Tool-execution vs. model-generation timing analysis:** We provide separate timing curves showing that tool costs remain comparable to model inference and decrease as policies improve.
  *(Requested by UHTW, see in section 8.)*

We also highlight that **VERLTOOL continues to evolve**: we have already updated the framework to the newest VERL version, expanded modality support (including audio), and improved compatibility with emerging RL algorithms such as SimpleTIR. We sincerely hope the AC will consider these substantial clarifications and updates into account. The detailed changelog is shown in Table 1:

Table 1: Summary of recent updates to VERLTOOL, including migration to the latest VERL version and addition of audio modality support.

| VERLTOOL Ver. | VERL Ver. | VLLM Ver. | Modality Support | Main Code Lines |
|---|---|---|---|---|
| 0.1.0 | 0.4.1 | 0.8.4 | Text, Image, Video | $\sim 1300$ |
| 0.2.0 | **0.6.0** | **0.11.0** | Text, Image, Video, **Audio** | $\sim 500$ |

Given that the reviewers are unable to engage due to recent ICLR review policy changes, we we sincerely AC could take a look of the rebuttal pdf in the supplementary material, where we have summarized all the rebuttal details in one place for you to read. Thank all the great suggestions raised by reviewers and valuable time from AC.

## 2 WHY PRIOR FRAMEWORKS CANNOT BE DIRECTLY EXTENDED TO VERLTOOL

We explain VERLTOOL's Distinctiveness as requested by reviewer `UHTW`, `YLgB`, and `PUT8`. To conclude, our framework addressed previous technical constraints, including **Tool Resource Management**, **Support for Stateful Tool Execution**, and **Broader Tool Feedback**. The table below clearly demonstrates our tool's broader coverage across all dimensions.

Table 2: Comparison of agentic reinforcement learning frameworks. The table shows key features including feedback format support, asynchronous rollout capabilities, tool execution characteristics, resource management systems, and vision-language model (VLM) compatibility. VERLTOOL (separated by dashed line) demonstrates comprehensive support across all evaluated dimensions.

| Agentic RL Frameworks | Tool/Env Feedback Format | Async Rollout | Independent Tool Server | Stateful Tool Execution? | Tool Resource Management | VLM Support? |
|---|---|---|---|---|---|---|
| OpenRLHF (gem) | Text | ✗ | ✓ | ✗ | - | ✗ |
| VeRL (MCP) | Text | ✓ | ✗ | ✗ | - | ✓ |
| ROLL (gem) | Text | ✗ | ✓ | ✗ | - | ✗ |
| RAGEN | Reward | ✗ | - | ✓ | - | ✗ |
| SLIME | Text | ✓ | ✗ | ✗ | - | ✗ |
| AREAL | Text | ✓ | ✗ | ✗ | - | ✓ |
| SKYRL | Text | ✓ | ✓ | ✓ | k8s | ✗ |
| Agent-Lightning | Text | ✓ | ✓ | ✗ | - | ✗ |
| VERLTOOL | Text/ Multi-modal/ Reward | ✓ | ✓ | ✓ | Ray | ✓ |

### 2.1 TECHNICAL CONSTRAINTS THAT PRIOR FRAMEWORKS ARE FACING

**Stateful Tool Execution.**  A key distinction is VERLTOOL's **per-trajectory stateful environment**. Existing frameworks (GEM, ROLL, OpenRLHF, SLIME) treat tools as stateless functions: each call is independent and cannot reference memory from previous steps. Agentic RL tasks such as multi-step code execution, iterative SQL debugging, or image refinement fundamentally require persistent state. VERLTOOL supports this through lightweight trajectory-scoped environments that maintain and update state across the entire rollout.

**Tool Resource Management.**  Many prior frameworks rely on **OS-level multiprocessing** for tool execution. This approach lacks flexible scheduling and offers no control over CPU/GPU contention across concurrent trajectories. Modern tools (Python interpreters, image/video operations, search retrievers) introduce substantial and heterogeneous resource demands. VERLTOOL manages these resources explicitly via **Ray actors**, enabling isolation, fault tolerance, batching, throttling, and cross-node scalability, while also controlling the lifecycle of per-trajectory stateful environments (creation, reuse, and cleanup) in a unified way.

**Diverse Forms of Tool Feedback.**  Unlike prior frameworks that restrict tools to text-only outputs, VERLTOOL supports **multimodal** feedback (images, videos, etc) and integrates these observations directly into the trajectory token stream. Moreover, the tool server can emit **action-level rewards** (e.g., execution correctness, SQL validity, retrieval quality) immediately upon each tool call, enabling fine-grained credit assignment and stable multi-turn optimization. This combination of multimodal observations and per-action reward signals is not supported in existing ARLT frameworks and is essential for complex, interleaved tool-use tasks.

### 2.2 CROSS-FRAMEWORK COMPARISON IN FEATURES

**Compared to VERL MCP.**  The MCP interface in VERL exposes tools through a **stateless JSON–RPC protocol** designed for single-turn interactions. It does not maintain trajectory-level

state (e.g., persistent Python variables, SQL cursors, image buffers), nor does it support multimodal outputs beyond plain JSON strings.

Simply adding more tools to MCP cannot replicate VERLTOOL's requirements of (1) stateful tool environments, (2) multimodal observation tokens.

**Compared to AREAL.** Reviewer 3 ("YLgB") raised a concern regarding the difference between VERLTOOL and AREAL. Here, we clarify our framework's uniqueness as follows:

i. AREAL provides an asynchronous RL training engine. However, offers **no unified tool-server abstraction**, no multimodal I/O format, and no mechanism for managing stateful tool calls. Moreover, it has **no well-designed tool-resource management strategies** for heterogeneous tools (e.g., Python sandbox, image processing, search retriever).

ii. Extending AREAL would require building precisely the components that VERLTOOL provides: (1) multimodal token handling, (2) a trajectory-scoped tool environment, and (3) a managed execution backend.

iii. VERLTOOL's ecosystem positioning is fundamentally different from AREAL and serves as a **complementary** role. Designed as a drop-in extension for the widely used veRL training stack, VERLTOOL's upstream alignment reduces the adoption cost and difficulties for researchers and machine learning engineers who are already familiar with VERL, and also ensures long-term compatibility as the VERL framework continues to evolve.

**Compared to SKYRL.** SKYRL integrates several task-specific tools, but its vision pipeline is **not a general multimodal API**, and cross-tool interactions are not unified under a single plugin interface. Its reliance on Kubernetes for resource management is **powerful but heavyweight**, making reproduction and extension difficult in typical research settings. Furthermore, tools are not exposed as a uniform abstraction, requiring substantial custom logic for each new tool type. VERLTOOL instead offers lightweight plugin definitions via `BaseTool`, unified across text, SQL, search, and multimodal tools.

## 2.3 Cross-framework Comparison in Runtime analysis under same workloads

As requested by reviewer UHTW, we evaluate the actual per-step training time in different frameworks under the same workload setting for Math-TIR framework. Results show VERLTOOLachieves the fastest per-step wall-clock time on 8 H100 GPUs.

Table 3: Wall-clock time analysis across different frameworks on Math-TIR tasks using the same setting. Each step samples 8 responses for 1024. The per-step time is averaged over the first 10 steps.

| FrameWork | per step time |
| --- | --- |
| SimpleTIR | 268 |
| verl-GEM | 206 |
| ROLL-GEM | - |
| AReal | 237 |
| VERLTOOL | **83** |

# 3 VERLTOOL'S ERROR HANDLING STRATEGIES DURING TOOL SERVER INTERACTION

Requested by reviewer UHTW and PUT8, here we provide more information about VERLTOOL's tool-server interaction error-handling strategies. In summary, VERLTOOL allows the **timeout** for tool execution on both the RL and the tool-server side, coupled with a **Toolcall Retrying Mechanism**. We provide two options: **Penalizing** and **Loss Marking** to deal with **erroneous trajectories**.

## 3.1 STEP-LEVEL TIMEOUT AND RETRYING MECHANISM

For tool-heavy tasks, each tool action is assigned a strict per-step timeout (e.g., 90 seconds in VT-SWE) to prevent slow or stalled tools from blocking long-horizon rollouts.

The tool server follows a *best-effort delivery* model: under heavy resource pressure (e.g., near-OOM), it may drop or refuse requests to stay alive. To distinguish between (1) a tool that is genuinely slow and (2) a request that was dropped due to overload, we use two timeout layers. The tool-server timeout is set slightly shorter than the trainer-side timeout, so tool-side delays trigger tool-server timeouts, while dropped requests appear as trainer-side timeouts. On the trainer side, users can adjust the retry behavior using the MAX_RETRY option, allowing the system to recover from transient failures.

To further stabilize training when resources are tight, VERLTOOL also allows users to limit the number of trajectories rolled out concurrently. This caps peak CPU/GPU/memory usage on the tool server and reduces the chance that retries fail due to resource exhaustion.

## 3.2 TRAJECTORY-LEVEL ERRONEOUS HANDLING

There are multiple ways to deal with erroneous trajectories. One simple way is to mask out the loss for those erroneous trajectories or those that do not produce a final answer. As shown in Figure 1, we implement Simple-TIR in VERLTOOL with minimal modification and get the same conclusion.

Similar strategy is also emploied in VT-SWE. As detailed in Appendix C.1 of the main paper, trajectories that encounter *tool errors*, *exceed length limits*, or *rolling time exceed 30 minutes* are removed from gradient computation using the same masking interface and are assigned a reward of zero under GRPO. This prevents corrupted or incomplete trajectories from contaminating credit assignment while preserving a clean learning signal for successful episodes.

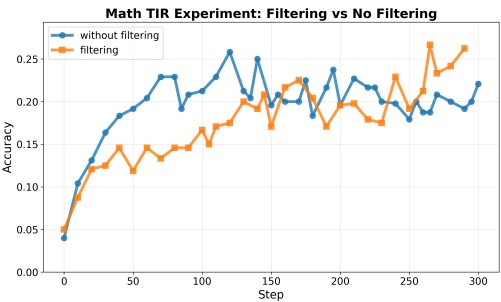
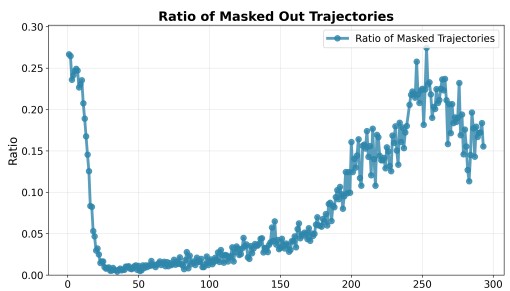

(a) The effects of filtering the erroneous trajectories or not on AIME-25 (avg@8) accuracy in Math-TIR experiments. Filtering trajectories like Simple-TIR (Xue et al., 2025) stabilizes the training.

(b) The ratio of masked out erroneous trajectories during training. The ratio starts high and decreases as the model improves, then increases to a stable 20%, which is the same observation as SimpleTIR.

Figure 1: Analysis of trajectory filtering in Math-TIR experiments. This not only demonstrates VERLTOOL is easy to modify and integrate more RL strategies like Simple-TIR, and proves that filtering the erroneous trajectories can stabilize the training.

## 4 TOOL SERVER IMPLEMENTATION DETAILS AND EFFICIENCY ANALYSIS

Following the request from reviewers `zwhJ`, `YLgB`, and `PUT8`, we detailed the tool server's implementation and performance. Our system **avoids all GIL bottlenecks** by design, achieving **low latency** for **light-weight tools** and **higher throughput** for **long-running tools**. Overall, the results show that our tool server is **efficient**, **scalable**, and suitable for **large-scale** agentic RL workloads.

### 4.1 TOOL SERVER DESIGN ISSUES AND OUR SOLUTIONS

Reviewer `YLgB` raised a concern regarding the potential impact of Python's GIL on multithreaded tool execution. We clarify that VERLTOOL's design is **not constrained by the GIL** due to deliberate separation of *request dispatch*, *process-based execution*, and *ray actor-based resource management*.

First, multithreading is used only for lightweight request dispatch. After dispatching, each tool-worker operates in one of two modes: (1) for resource-intensive tools, the worker launches a new OS process via `subprocess`, fully bypassing the GIL (e.g., `Python Interpreter`, `Image Processing`); (2) for lightweight, I/O-bound tools, execution is handled via `asyncio`, which does not block the GIL (e.g., `search-retriever`, `Google-Search`).

Second, many heavier tools are managed as **Ray Actors**, each running in an isolated actor process with explicitly allocated CPU/GPU resources. These processes never share a Python interpreter or GIL, offering robust parallelism and fine-grained resource control.

The execution model is summarized below:

| Tool Resource Type | Server Mode | Execution Backend | GIL Impact |
|---|---|---|---|
| Resource-intensive (CPU/GPU) | Threading
Ray Actor | OS process via `subprocess`
ray process with resources | None (new process)
None (actor process) |
| I/O-bound, lightweight | Threading
Ray Actor | `asyncio` event loop
ray process with resources | None (non-blocking)
None (actor process) |

Table 4: Execution paths of VERLTOOL's tool server and their interaction with the Python GIL.

### 4.2 TOOL SERVER EFFICIENCY ANALYSIS

As requested by Reviewer `PUT8`, we conducted a detailed efficiency analysis of the tool-server backend. Across both lightweight and realistic workloads, the results show that VERLTOOL's tool server maintains *stable latency*, *high throughput*, and *graceful degradation* under increasing load.

For lightweight Python calls (Table 5), the server achieves an average response time of **1.04s** at its optimal configuration (1024 max concurrency, 256 request concurrency), and remains stable even under extreme stress (4096 concurrent requests) with a bounded average latency of **9.1s**.

For realistic long-running tasks with randomized 1–10s execution time (Table 6), the tool server matches baseline performance at moderate concurrency and achieves a **9.7% higher throughput** at high concurrency (1024), confirming that kernel pooling and process-based isolation efficiently handle heterogeneous, long-duration tool calls. These measurements validate the practicality of VERLTOOL's resource-managed tool execution design and its ability to support large-scale, highly parallel agentic RL workloads.

### 4.3 HOW DOES VERLTOOL ISOLATE TOOL ENVIRONMENTS?

**Virtualization and Isolation.** Verl-Tool adopts a *layered virtualization strategy* tailored to the security and performance needs of each tool type. Instead of relying on a single mechanism, the framework combines lightweight sandboxes for high-throughput RL workloads with heavier OS-level environments when full system access is necessary.

**Code-execution tools** (e.g., Python environments used in Math-TIR or DeepSearch) run inside *fire-jail* sandboxes, providing process and filesystem isolation with negligible overhead. **Browser-based**

Table 5: Performance comparison for lightweight Python execution (`print("hello world")`). The tool-server demonstrates efficient handling of simple operations across varying concurrency levels. At the optimal configuration (1024 max concurrency, 256 request concurrency), the average response time is **1.04s** with maximum latency of only 1.43s, showing excellent scalability. Performance degrades gracefully under extreme load (4096 requests) but remains stable with 9.1s average response time.

| Mode | Tool-Server Max Concurrency | Request Concurrency | Response Time (s) | | | Note |
|------|------|------|------|------|------|------|
| | | | Min | Avg | Max | |
| | 128 | 256 | 0.29 | 3.47 | 20.00 | Under-provisioned |
| | 512 | 256 | 0.38 | 1.25 | 2.15 | Good |
| | 1024 | 256 | 0.36 | **1.04** | 1.43 | **Optimal** |
| **Ray** | 4096 | 256 | 0.48 | 1.15 | 1.44 | Over-provisioned |
| | 4096 | 512 | 0.28 | 1.46 | 2.20 | Scaling up |
| | 4096 | 1024 | 0.26 | 2.81 | 3.99 | Heavy load |
| | 4096 | 2048 | 0.41 | 5.81 | 8.67 | Very heavy load |
| | 4096 | 4096 | 0.47 | 9.10 | 16.03 | Extreme load |

Table 6: Performance comparison for realistic computation simulation using `import time; import random; time.sleep(random.randint(1,10)); print('Hello from stress test!')`. This benchmark simulates real-world scenarios with variable execution time (1-10s random sleep). **Key Finding**: The tool-server (VERLTOOL Server) achieves **9.7% higher throughput** (83.11 vs 75.79 req/s) at high concurrency (1024) while maintaining comparable latency. At moderate concurrency (256), both approaches perform similarly, indicating that the tool-server's benefits become pronounced under higher load conditions. The tool-server's kernel pooling approach efficiently manages concurrent long-running tasks without significant overhead.

| Backend | Max Concurrency | Request Concurrency | Response Time (s) | | | Throughput (req/s) |
|------|------|------|------|------|------|------|
| | | | Min | Avg | Max | |
| Baseline | 256 | 256 | 1.29 | 6.01 | 10.63 | 23.54 |
| VERLTOOL Server | 256 | 256 | 1.16 | 6.23 | 11.64 | 21.54 |
| Baseline | 1024 | 1024 | 1.30 | 6.69 | 11.96 | 75.79 |
| VERLTOOL Server | 1024 | 1024 | 1.22 | 7.31 | 11.18 | **83.11** |

**tools** use Ray-managed Playwright sessions, where each request is served in an isolated browser context to avoid cross-trajectory interference and ensure deterministic behavior. For **software engineering tasks**, Verl-Tool employs *Docker-based SWE environments* that support persistent per-trajectory state and strong OS-level isolation, similar to real-world development setups.

In the SWE Docker environment, *sudo* can be enabled safely due to container-level isolation; however, our tasks do not require elevated privileges. Verl-Tool follows a strict *least-privilege* policy, allocating stronger permissions only when demanded by task semantics. For typical ARLT tasks, unprivileged containers and firejail sandboxes are sufficient, ensuring both safety and training efficiency.

Overall, this multi-layer design achieves (i) scalable lightweight isolation, (ii) strong OS-level virtualization when needed, and (iii) a unified interface for tool developers without exposing host-level privileges.

## 5 ABLATION ON TOKENIZATION STRATEGIES

As requested by Reviewers `UHTW`, `zwhJ`, and `PUT8`, we provide additional evidence on why the tokenization strategy used in VERLTOOL—specifically, avoiding retokenizing model outputs—is important for stable agentic RL. We show that (1) concurrent work has independently validated the same issue, and (2) our own experiments confirm that retokenization significantly affects both training stability and downstream performance.

### 5.1 CONCURRENT WORK FROM VLLM CONFIRMS THE IMPORTANCE OF "NO RETOKENIZATION"

Shortly after our submission, vLLM published a detailed analysis ("No More Retokenization Drift: Returning Token IDs via the OpenAI Compatible API Matters in Agent RL", Oct 22, 2025) highlighting two key issues: (1) the same string can produce *different token ID sequences* when retokenized, and (2) this retokenization drift can cause *rapid collapse* in agentic RL training. Figure 2 summarizes their findings.

These observations are fully consistent with what we raised in our paper (Figure 3), despite being discovered independently and after our ICLR submission. This strengthens the significance and correctness of the issue we identified.

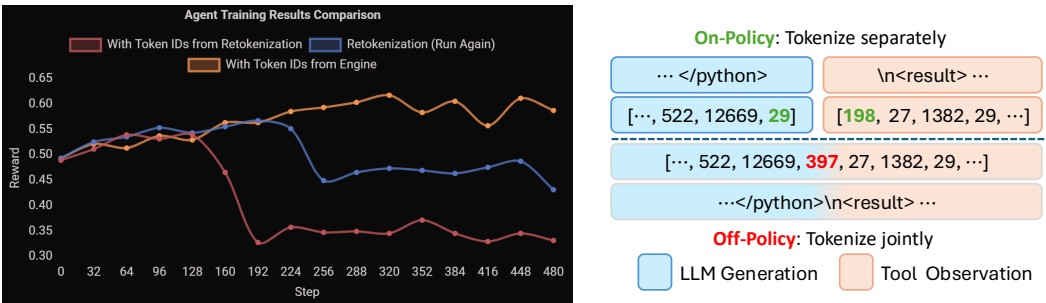

Figure 2: Retokenization drift analysis from the Agent Lightning blog.

Figure 3: Retokenization boundary issue highlighted in our paper.

### 5.2 OUR ADDITIONAL EXPERIMENTS VALIDATING THE SAME PHENOMENON

To directly measure the impact of retokenization on agentic RL using VERLTOOL framework, we conducted additional experiments on the SQL task (Spider). The results in Figure 4 show three effects:

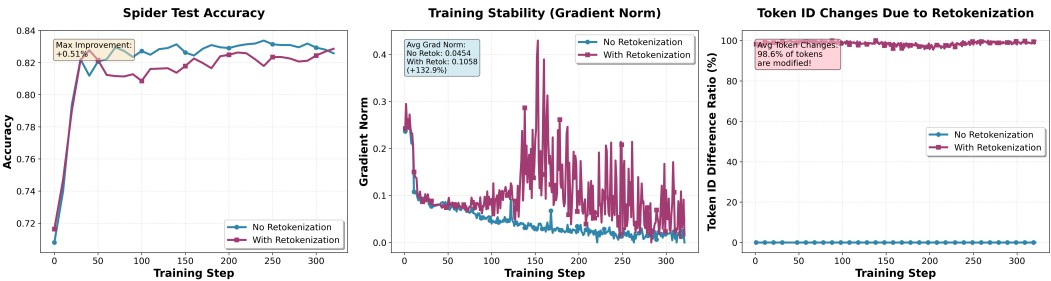

Figure 4: Ablation of tokenization strategies in our SQL agentic RL experiment. Retokenization causes instability and performance degradation.

- **(1) Performance drift:** The retokenized model exhibits inconsistent or degrading reward dynamics, matching the collapse behavior noted by vLLM. In contrast, using the original token IDs provided by the model yields smooth and monotonic improvement.

- **(2) Training instability:** Retokenization inflates the gradient norm by **132.9%** on average, indicating unstable policy updates.
- **(3) Large token-ID mismatch:** Retokenization modifies **98.6%** of token IDs during training, demonstrating that token boundary inconsistencies are pervasive and not rare corner cases.

Together, these results show that retokenization does not merely create cosmetic differences—it substantially alters the underlying training trajectory and destabilizes policy optimization.

# 6 USE VERLTOOL TO TRAIN A SINGLE MODEL WITH MULTIPLE TOOL TYPES.

Following the requests from reviewers `YLGB` and `UHTW`, we highlight that VERLTOOL can train a single model to use multiple tool types within the same trajectory. Such multi-tool settings naturally arise in real-world agentic tasks and remain scalable under long-horizon training.

## 6.1 EXISTING MULTI-TOOL RL EXPERIMENTS IN THE PAPER

**VT-DeepSearch** VT-DeepSearch combines the `Python Interpreter` and `Web Search` within a single rollout. As shown in Table 7 of the paper, it outperforms standard search-only baselines (e.g., WebThinker, Search-o1) on both GAIA and HLE. This demonstrates that heterogeneous tool use remains stable and can yield stronger retrieval-driven reasoning than single-tool setups.

**VT-SWE.** VT-SWE integrates `bash commands`, `file I/O`, and `search tools` within the same rollout. The training trajectory involves up to 50 interaction turns under a 32K context window. Using SWE-Verified as an example, Table 7 summarizes the distribution and runtime characteristics of tools observed.

| Tool Name | Avg. Count | Avg. Turns | Avg. Time (s) | P99 Time (s) |
|---|---|---|---|---|
| `file:view` | $17.15 \pm 6.21$ | $21.54 \pm 17.79$ | $0.3120 \pm 1.3912$ | 1.0026 |
| `bash` | $15.78 \pm 6.30$ | $35.06 \pm 16.24$ | $1.2191 \pm 6.1584$ | 11.6799 |
| `search` | $9.88 \pm 6.01$ | $24.36 \pm 17.07$ | $0.2362 \pm 0.2002$ | 0.5808 |
| `file:replace` | $6.76 \pm 3.51$ | $32.08 \pm 15.98$ | $0.3086 \pm 1.5522$ | 0.8457 |
| `file:create` | $4.75 \pm 2.47$ | $28.81 \pm 16.87$ | $0.2715 \pm 0.2230$ | 0.6098 |

Table 7: Tool distribution for VT-SWE, evaluation on SWE-Verified, maximum 100 rounds.

## 6.2 NEWLY ADDED MULTI-TOOL RL EXPERIMENTS.

Still, as requested by reviewer `zwhJ` and `YLgB`, we have conducted some multi-tool RL experiments that training a single model with different types of tools at the same time. We combine Math-TIR with SQL-Coder as Math-SQL multi-tool experiments. And combine Math and Visual Reasoning as Math-Visual Training. Although we have tried to train a model with more than 2 types of tools and tasks at a time, but all the results turn to that the training collapsed. We summarize the reason in subsection 6.3.

Table 8: Copmarison of performance on training 290 steps. The single-tool denotes the Qwen2.5-Coder-Instruct training for Spider-relistic and Qwen-2.5-Math for AIME. The multi-tool experiment trains Qwen2.5-Coder-Instruct on the SQL-Coder and Math-TIR joints.

| Method (290 steps) | Spider-realistic | AIME 24 | AIME 25 |
|---|---|---|---|
| Single-Tool | 81.3 | **28.75** | 19.17 |
| Multi-Tool | **81.5** | 23.33 | **26.67** |

Table 9: Copmarison of performance on training 290 steps. The single-tool denotes the Qwen2.5-Coder-Instruct training for Spider-relistic and Qwen-2.5-Math for AIME. The multi-tool experiment trains Qwen2.5-Coder-Instruct on the SQL-Coder and Math-TIR joints.

| Method (80 steps) | V-Star | AIME 24 | AIME 25 |
|---|---|---|---|
| Single-Tool | **81.3** | **28.75** | **19.17** |
| Multi-Tool | 59.0 | 0 | 3 |

**Math-SQL**    We first train Qwen-2.5-Coder-Instruct-7B on Math-TIR and SQL Coder tasks jointly. Results are shown in Table 8 and Figure 5. We see a joint benefit on downstream task performance due to the similarity of Math and SQL task.

**Math-Visual**    We then train Pixel-Reaoner-7b-WarmStart on Math-TIR and Visual-Reasoner tasks jointly. Results are shown in Table 9 and Figure 6. However, the joint multi-tool training don't show enough benefit for either task.

## 6.3    WHY SIMPLY COMBINING MULTI-TOOL TRAINING DOES NOT WORK

We attribute the collapse of models trained with three or more tool types to two factors: (1) the lack of an effective cold-start phase, which leads the policy to explore invalid tool–task combinations early in training, and (2) conflicting optimization objectives across heterogeneous tasks, which produce unstable gradients and mutually interfering reward signals. While single-tool and closely related two-tool settings (e.g., Math-SQL) can share representations, broader multi-tool mixtures require additional prior knowledge, tool-specific warmup stages, or hierarchical routing to avoid destructive interference. We leave these techniques for future work.

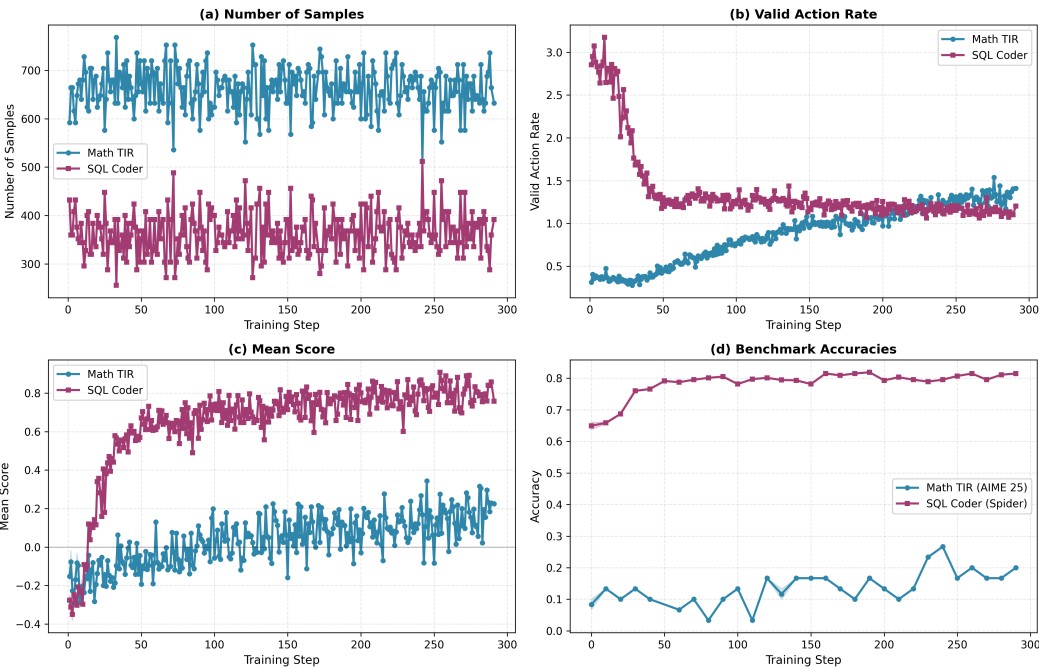

Figure 5: The joint training metrics on Qwen-2.5-Coder-Instruct-7B using two types of tools and tasks at the same time. Joint benefit shown on both tasks.

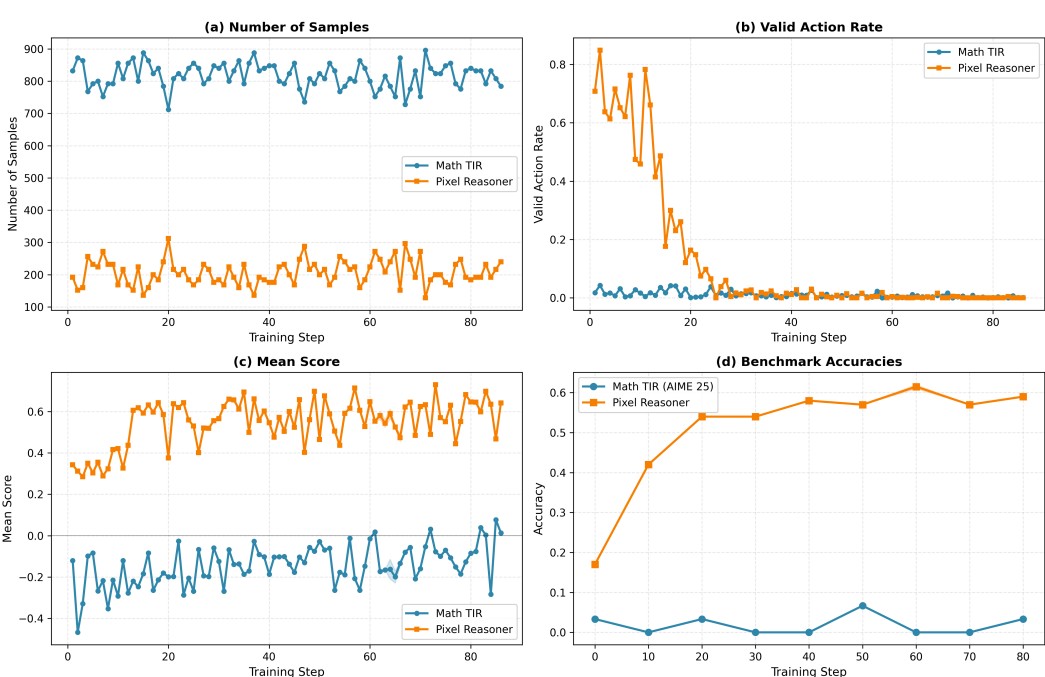

Figure 6: The joint training metrics on Pixel-Reaoner-7b-WarmStart checkpoint using two types of tools and tasks at the same time. No joint benefit shown on either task.

# 7 CLARIFYING OCCASIONAL VT-* IMPROVEMENTS OVER BASELINES

As requested by reviewer UHTW, we clarify the reason behind the performance differences between our agents and baselines. We conclude that this is mainly due to differences in **Training Parameter Settings**, **Tool-Calling Formats**, and **Maximum Response Length**, with detailed analysis.

## 7.1 PARAMETER DIFFERENCES

Parameter difference can have a noticeable impact on agents' performance. For VT-Search-zero, our training parameter differs from the original Search-R1 in the following key aspects. Firstly, we set the maximum tool response length to $2048$ while the original Search-R1 uses $500$. As discussed below, this parameter has a significant impact on the performance of local dense retrieval agents. Additionally, we set the group size in GRPO as $16$ while Search-R1 is $5$. For VT-VisualReasoner, it falls slightly short of the PixelReasoner, largely because our agent was trained for only 50 steps, whereas the official baseline is obtained from a full 400-step model.

Table 10: Key parameter differences between our VT-* agents and the original baselines.

| Agent | Parameter | Ours | Original Baseline |
|---|---|---|---|
| VT-Search-zero | Max tool response length $l$ | 2048 | 500 |
| | GRPO group size | 16 | 5 |
| VT-VisualReasoner | Training steps | 50 | 400 |

## 7.2 SCALING EFFECTS ANALYSIS ON THE MAXIMUM RESPONSE LENGTH

For context-dependent tasks, scaling the response length can significantly impact the agent's performance. For the Search-R1 (local dense retrieval) task, we scaled the maximum tool response length $l$ from $512$ to $4096$. As shown in the table below, setting $l$ to $512$ yields suboptimal performance, whereas scaling to $1024$ and $2048$ results in noticeable performance gains. However, scaling further has little improvement in model performance. This is because shortening $l$ risks truncating the information that models could use to answer questions correctly. On the other hand, as the returned documents are ranked by semantic relevance, including less related documents may distract the model and hinder it from identifying the most crucial clue.

Table 11: Effect of scaling the maximum tool response length $l$ on Search-R1 Task, best-performing metrics are highlighted in bold.

| Benchmarks | $l$=512 | $l$=1024 | $l$=2048 | $l$=4096 |
|---|---|---|---|---|
| NQ | 0.234 | **0.475** | 0.473 | 0.469 |
| TriviaQA | 0.445 | 0.634 | **0.637** | 0.628 |
| PopQA | 0.179 | 0.469 | 0.485 | **0.488** |
| HotpotQA | 0.206 | 0.351 | 0.348 | **0.352** |
| 2Wiki | 0.259 | 0.324 | 0.331 | **0.334** |
| Musique | 0.043 | 0.090 | 0.087 | **0.093** |
| Bamboogle | 0.088 | **0.232** | 0.200 | 0.192 |

# 8 TOOL EXECUTION METRICS

As requested by reviewer `UHTW`, we conducted a detailed breakdown of time spent on model inference versus tool execution. As shown in Figure 7, tool-call latency initially rises due to a large number of inefficienct code written by the model, then rapidly decreases as the policy learns to issue more well-formed tool calls (middle plot). After stabilization, tool execution and model-generation times remain comparable throughout training. This confirms that (1) tool use does not dominate end-to-end cost, (2) asynchronous execution amortizes tool latency effectively, and (3) model improvements directly translate into more efficient tool usage and higher final accuracy on AIME (right plot).

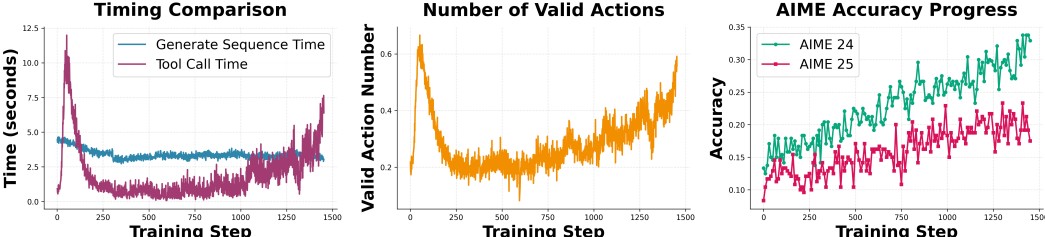

Figure 7: Training metrics for Math-TIR tasks on Qwen-1.5B-Math model

# REFERENCES

Zhenghai Xue, Longtao Zheng, Qian Liu, Yingru Li, Zejun Ma, and Bo An. Simpletir: End-to-end reinforcement learning for multi-turn tool-integrated reasoning. https://simpletir.notion.site/report, 2025. Notion Blog.