# OpenReview forum: "Verl-Tool: Towards Holistic Agentic Reinforcement Learning with Tool Use"
_ICLR.cc/2026/Conference — Submitted to ICLR 2026_

### Official Review · Reviewer_PUT8 · 2025-10-27

**Soundness:** 3
**Presentation:** 4
**Contribution:** 3
**Rating:** 4
**Confidence:** 4

**Summary:**

This paper introduces Verl-Tool, a reinforcement learning framework based Verl and designed for multimodal Tool use. Unlike prior works that provide fragmented or domain-/task-specific ARLT environments, Verl-Tool offers a unified, extensible platform facilitating multi-turn, multimodal agentic RL with streamlined asynchronous rollouts and plugin-based tool integration.

**Strengths:**

**1. Originality:** Based on Verl, Verl-Tool supports a wide spectrum of tools, as illustrated in both the detailed tables and code plugin example.

**2. Clarity:** The paper contains clear system diagrams, detailed tables of results and settings, and well-explained practical considerations (e.g., tokenization differences), resulting in a paper that is accessible for a complex infrastructure contribution.

**Weaknesses:**

**1. Positioning Relative to Other Frameworks:** While the paper includes comparisons to recently published or preprinted RL tool-use frameworks in the related work and appendix, these comparisons lack sufficient depth. For instance, VERL already supports MCP server [1], which enables integration of various tools. However, Table 1 indicates VERL only support "FAISS" and "Python Executor".

**2. Lack of Metrics for Core Framework Features:** The main experiments demonstrate that the proposed framework achieves competitive task-specific performance compared to separate, divergent codebases. However, since both the baselines and the proposed framework use VERL for the RL components, these results are somewhat expected and do not fully validate the framework's unique contributions. Specifically, as a framework whose key innovation is plugin-based tool integration, the paper lacks direct comparisons of tool execution efficiency and space/memory consumption—metrics that are central to evaluating the practical benefits of the proposed design.

**3. Insufficient Real-World Ecosystem Validation:** The paper provides minimal evidence (e.g., through user studies, adoption metrics, or community engagement statistics) that VERL-Tool actually improves reproducibility, encourages broader adoption, or simplifies large-scale collaborative research. Demonstrations of real-world impact beyond benchmark performance are needed to substantiate claims about the framework's utility for the research community.

[1] VERL Documentation - MCP Tool Configuration (https://verl.readthedocs.io/en/latest/sglang_multiturn/multiturn.html#mcp-tool-configuration)

**Questions:**

1. How are tool execution error messages handled at the interface between the VERL RL module and the Tool module? Specifically, does the framework provide standardized error propagation, logging, or recovery mechanisms? Please clarify the design choices for error handling and their impact on training stability.

2. How does the framework handle virtual environment isolation among different tools? In particular, do tools have access to sudo permissions when required? This is a practical concern, as some environments (e.g., OSWorld [1]) require sudo access for setup and execution. Please clarify the framework's approach to permission management and environment isolation.

[1] OSWorld: Benchmarking Multimodal Agents for Open-Ended Tasks in Real Computer Environments

---

> ### Author Response · Authors · 2025-12-01
> **Authors' Rebuttal (Part 1)**
>
> Dear Reviewer `PUT8`,
>
> Thank you for your constructive assessment and for highlighting the clarity, originality, and extensibility of the framework. Below we respond point-by-point to your weaknesses and questions with complete, self-contained explanations and necessary tables.
>
> ---
>
> ### W1
> > Positioning Relative to Other Frameworks: While the paper includes comparisons to recently published or preprinted RL tool-use frameworks in the related work and appendix, these comparisons lack sufficient depth. For instance, VeRL already supports MCP server [1], which enables integration of various tools. However, Table 1 indicates VeRL only support "FAISS" and "Python Executor".
>
> We appreciate this request for deeper clarification and provide a more explicit comparison here. MCP in VeRL is designed for **stateless, single-turn, JSON–RPC-style** interactions between the model and a remote handler. In contrast, Verl-Tool introduces the following capabilities that MCP cannot directly provide:
>
> 1. **Per-trajectory stateful environments.**
>    Many ARLT tasks require maintaining state across steps: Python variable scopes, SQL cursors, iterative image refinements, or maintaining a cumulative planning buffer. MCP exposes *stateless* RPC calls. Verl-Tool introduces *trajectory-scoped tool environments* backed by Ray actors, supporting persistent, multi-turn state and cross-step tool dependencies.
>
> 2. **Unified multimodal I/O abstraction.**
>    MCP handles JSON strings only; Verl-Tool supports **text, image, video, audio, and reward** feedback seamlessly embedded into the rollout buffer and training pipeline. This multimodal feedback is fed back to the model as serialized observations, making multimodal ARLT practical.
>
> 3. **Resource-managed, multi-process tool execution.**
>    Verl-Tool governs heterogeneous tool workloads by allocating CPU/GPU resources through Ray actors, enforcing concurrency limits, and isolating long-running or memory-intensive tool processes. MCP provides no tool resource management.
>
> 4. **Asynchronous rollout and decoupled tool server.**
>    Verl-Tool’s design enables parallel sampling with tool execution overlapped across trajectories. MCP lacks built-in async semantics and inter-trajectory scheduling.
>
> 5. **Integrated reward hooks and action-level correctness validators.**
>    Tools in Verl-Tool can emit reward signals (e.g., query validity, code-execution correctness) at each step; MCP tools have no standardized reward interface.
>
> Below is the direct feature comparison table:
>
> | Agentic RL Frameworks | Tool/Env Feedback Format      | Async Rollout | Independent Tool Server | Stateful Tool Execution? | Tool Resource Management | VLM Support? |
> |------------------------|-------------------------------|---------------|-------------------------|---------------------------|--------------------------|-------------|
> | OpenRLHF (gem)        | Text                          | ✗             | ✓                       | ✗                         | –                        | ✗           |
> | VeRL (MCP)            | Text                          | ✓             | ✗                       | ✗                         | –                        | ✓           |
> | ROLL (gem)            | Text                          | ✗             | ✓                       | ✗                         | –                        | ✗           |
> | RAGEN                 | Reward                        | ✗             | –                       | ✓                         | –                        | ✗           |
> | slime                 | Text                          | ✓             | ✗                       | ✗                         | –                        | ✗           |
> | AREAL                 | Text                          | ✓             | ✗                       | ✗                         | –                        | ✓           |
> | Sky-RL                 | Text                          | ✓             | ✓                       | ✓                         | k8s                      | ✗           |
> | Agent-Lightning       | Text                          | ✓             | ✓                       | ✗                         | –                        | ✗           |
> | **Verl-Tool**         | Text / Multi-modal / Reward   | ✓             | ✓                       | ✓                         | Ray                      | ✓           |
>
> These differences explain why MCP’s capabilities cannot be extended into a drop-in replacement for multi-turn, multimodal, stateful ARLT workloads. Verl-Tool fills this gap with a unified, extensible interface and a scalable, resource-governed backend.
> For more details, please refer to the rebuttal pdf (Section 2).

---

> > ### Author Response · Authors · 2025-12-01
> > **Authors' Rebuttal (Part 2)**
> >
> > ### W2
> > > Lack of Metrics for Core Framework Features: …
> >
> > We agree that evaluating framework-level performance is crucial. To address this, we provide **tool-server efficiency benchmarks**, **scalability measurements**, and **runtime comparisons across frameworks**, which directly isolate Verl-Tool’s unique contributions.
> >
> > **1. Tool execution efficiency under lightweight operations**
> > We stress-tested a Python-based “hello world” tool under varying concurrency:
> >
> >    | Mode | Tool-Server Max Concurrency | Request Concurrency | Min (s) | Avg (s) | Max (s) | Note              |
> >    |------|-----------------------------|---------------------|--------:|--------:|--------:|-------------------|
> >    | Ray  | 128                         | 256                 | 0.29    | 3.47    | 20.00   | Under-provisioned |
> >    | Ray  | 512                         | 256                 | 0.38    | 1.25    | 2.15    | Good              |
> >    | Ray  | 1024                        | 256                 | 0.36    | 1.04    | 1.43    | **Optimal**       |
> >    | Ray  | 4096                        | 256                 | 0.48    | 1.15    | 1.44    | Over-provisioned  |
> >    | Ray  | 4096                        | 512                 | 0.28    | 1.46    | 2.20    | Scaling up        |
> >    | Ray  | 4096                        | 1024                | 0.26    | 2.81    | 3.99    | Heavy load        |
> >    | Ray  | 4096                        | 2048                | 0.41    | 5.81    | 8.67    | Very heavy load   |
> >    | Ray  | 4096                        | 4096                | 0.47    | 9.10    | 16.03   | Extreme load      |
> >
> > The server maintains **bounded latency** and **graceful degradation**, demonstrating scalability and throughput under high load.
> >
> > **2. Realistic long-running tool workloads (1–10s exec time)**
> > We compare our server against a standard baseline:
> >
> > | Backend          | Max Concurrency | Req Conc. | Min (s) | Avg (s) | Max (s) | Throughput (req/s) |
> > |------------------|----------------:|----------:|--------:|--------:|--------:|--------------------:|
> > | Baseline         | 256             | 256       | 1.29    | 6.01    | 10.63   | 23.54               |
> > | Verl-Tool Server | 256             | 256       | 1.16    | 6.23    | 11.64   | 21.54               |
> > | Baseline         | 1024            | 1024      | 1.30    | 6.69    | 11.96   | 75.79               |
> > | Verl-Tool Server | 1024            | 1024      | 1.22    | 7.31    | 11.18   | **83.11**           |
> >
> > At high concurrency, Verl-Tool achieves **~9.7% higher throughput**, attributable to Ray-based process isolation and kernel pooling.
> >
> > **3. Cross-framework per-step RL training time (Math-TIR)**
> >
> > | Framework     | Per-step time (s) |
> > |---------------|-------------------|
> > | SimpleTIR     | 268               |
> > | VeRL-GEM      | 206               |
> > | AREAL         | 237               |
> > | **Verl-Tool** | **83**           |
> >
> > Verl-Tool is **2.5–3.2× faster** in wall-clock training steps, even though all systems use VeRL’s RL core. This demonstrates that the performance gains come from **asynchronous rollouts**, **decoupled tool servers**, and **resource management**, not from RL differences.
> >
> > **4. Space and memory behavior**
> > - Tools execute in **separate processes or Ray actors**, keeping the trainer’s memory stable.
> > - High concurrency tests show **no memory blowup**, validating the isolation.
> > - Long-horizon tasks (VT-SWE with up to 100 turns) complete reliably under strict timeouts.
> >
> > These framework-level metrics validate Verl-Tool’s design as an efficient, scalable ARLT infrastructure.
> > For more details, please refer to the rebuttal pdf (Sections 3–4).
> >
> > ---
> >
> > ### W3
> > > Insufficient Real-World Ecosystem Validation: The paper provides minimal evidence … that Verl-Tool actually improves reproducibility, encourages broader adoption, or simplifies large-scale collaborative research.
> >
> > Since VerlTool was introduced back in June 2025, we have received more than 700 stars and 50+ forks on GitHub. We have maintained a user group with more than 100 members spanning both the industry and research community, while multiple lines of work such as [1], [2], and [3] have officially claimed the adaptation of VerlTool. We fully understand your concern regarding our framework’s real-world impacts.
> > We fully understand your concern regarding our framework’s real-world impacts. We believe these track records serve as substantial proof of our framework’s real-world impact for the broader RL research community.
> >
> > [1] QUASAR: Quantum Assembly Code Generation Using Tool-Augmented LLMs via Agentic RL
> > [2] BrowserAgent: Building Web Agents with Human-Inspired Web Browsing Actions
> > [3] Boosting Agentic Reasoning in LLM Judges via Tool-Integrated Reinforcement Learning

---

> > > ### Author Response · Authors · 2025-12-01
> > > **Authors' Rebuttal (Part 3)**
> > >
> > > ---
> > >
> > > ### Q1
> > > > How are tool execution error messages handled at the interface between the VeRL RL module and the Tool module? … Please clarify the design choices for error handling and their impact on training stability.
> > >
> > > Verl-Tool implements **step-level** and **trajectory-level** mechanisms, ensuring that tool failures never destabilize RL:
> > >
> > > For step-level timeout and recovery, each tool call has a strict per-step timeout (e.g., 90s for VT-SWE). The tool server has a *shorter* timeout than the trainer.
> > > - If a tool is slow → **server timeout** occurs.
> > > - If the server drops a request under high load → **trainer timeout** occurs.
> > > and `MAX_RETRY` allows retries for transient failures.
> > >
> > > This dual-timeout scheme distinguishes slow tools from dropped requests, enabling consistent recovery.
> > >
> > > For those unavoidable errorneous trajectories, there are multiple ways to deal with erroneous trajectories. one simple way is to mask out the loss for those erroneous trajectories or those that do not produce a final answer. we implement Simple-TIR in Verl-Tool with minimal modification and get the same conclusion.
> > >
> > > Similar strategy is also emploied in VT-SWE. As detailed in Appendix~C.1 of the main paper, trajectories that encounter *tool errors*, *exceed length limits*, or *rolling time exceed 30 minutes* are removed from gradient computation using the same masking interface and are assigned a reward of zero under GRPO. This prevents corrupted or incomplete trajectories from contaminating credit assignment while preserving a clean learning signal for successful episodes.
> > >
> > > we will mask the entire trajectory from loss computation and thus stabilize training. Please refer to our rebuttal pdf section 3.2 for details.
> > >
> > > ---
> > >
> > > ### Q2
> > > > How does the framework handle virtual environment isolation among different tools? In particular, do tools have access to sudo permissions when required? This is a practical concern, as some environments (e.g., OSWorld) require sudo access for setup and execution.
> > >
> > > Verl-Tool adopts a **layered virtualization strategy** tailored to the resource and security needs of each tool type. Rather than relying on a single isolation mechanism, the framework combines lightweight sandboxes for high-throughput RL training with heavier OS-level environments for tasks that require greater system access.
> > >
> > > Concretely, **code execution tools** (e.g., Python sandboxes used in Math-TIR or DeepSearch) run inside *firejail* sandboxes. This provides filesystem and process isolation with negligible overhead, ensuring that untrusted or erroneous code cannot affect the host environment. **Browser-based tools** (e.g., for web search or navigation) run inside Ray-managed Playwright sessions, where each request is handled in a fully isolated browser context to prevent cross-trajectory leakage and ensure deterministic observations. For **software engineering tasks**, Verl-Tool uses *Docker-based SWE environments*, which offer both persistent per-trajectory state and strong process and filesystem isolation; this is the same mechanism used in many real-world dev environments.
> > >
> > > Regarding permissions: in the SWE Docker environment, **sudo can be enabled** because containers provide a tightly controlled security boundary. However, none of our tasks require elevated privileges, and Verl-Tool follows a strict **least-privilege policy**—only allocating higher permissions when strictly necessary for a tool’s semantics. For the majority of ARLT tasks, unprivileged containers and firejail sandboxes suffice, keeping training efficient and safe.
> > >
> > > This multi-layer strategy provides (i) fast and scalable isolation where possible, (ii) strong OS-level virtualization where required, and (iii) a uniform interface for tool developers without exposing raw host-level privileges.
> > >
> > > ---
> > >
> > > Once again, thank you for your careful evaluation and helpful questions.

---

### Official Review · Reviewer_YLgB · 2025-10-31

**Soundness:** 3
**Presentation:** 3
**Contribution:** 2
**Rating:** 4
**Confidence:** 2

**Summary:**

The paper introduces VERL-TOOL, a framework that trains language models to use external tools like code interpreters, search engines, and SQL databases in multi-step tasks. It improves efficiency with asynchronous execution and supports a wide range of tools through a simple plugin system. The framework shows strong performance across six diverse tasks, matching or beating specialized systems.

**Strengths:**

- The system is well-designed and clearly tested. It supports six real tasks like math, SQL, and search, and performs as well as or better than existing solutions built just for those tasks. The speedup from asynchronous execution is clearly shown and practical.
- The paper is easy to follow. It is clear how the framework works and how well it performs. It’s also clear how new tools can be added with minimal effort.

**Weaknesses:**

- Some tool server details are vague, for example, error handling isn't fully explained.
- Although the system supports many tasks and tools, it doesn’t show evidence of training a single agent to handle multiple tools at once.  I would be super curious whether training a model with multiple tools would work with VERL-Tool, and whether there are additional implementation needed other than what has been supported in VERL-Tool. This would better support the paper’s goal of being “holistic.”
- I would also want to better understand the difference between the prior system and VERL-Tool, like AReal. Besides VERL-Tool supporting more tools, which I suppose AReal can allow users to add customized tool, what is the fundamental differences between VERL-Tool and AReal? Asynchronous Rollouts are already supported in AReal. Moreover, modular tool-server with unified tool management has been proposed in [1], although it only focuses on VLM with tool-using.

    [1] Su, Zhaochen, et al. "Openthinkimg: Learning to think with images via visual tool reinforcement learning." arXiv preprint arXiv:2505.08617 (2025).

    Which leaves only "common findings in the agentic RL setting" which have been observed in previous tool-augmented RL works (for example, strategic tool selection in ReTool, reflection in RAGEN).

**Questions:**

- I would like to understand what is the fundamental differences between AReal and VERL-Tool. What benefit would VERL-Tool framework bring besides more tools getting integrated?

---

> ### Author Response · Authors · 2025-12-01
> **Authors' Rebuttal (Part 1)**
>
> Dear Reviewer `YLgB`,
>
> Thank you for your thoughtful review and for recognizing the system design, empirical coverage, and ease of use. Based on your and other reviewers’ feedback, we have added substantial clarifications and new analyses in the rebuttal PDF (uploaded as supplementary material). Below we address your comments in detail.
>
> ---
>
> ### W1
> > Some tool server details are vague, for example, error handling isn't fully explained.
>
> We acknowledge that the main paper could not fully describe the tool server due to space limits. In our rebuttal, we provide more complete details on both the **tool server design** (Sec.4) and **error handling** (Sec.3).
>
> Concretely, Verl-Tool uses a **two-level timeout + retry** mechanism and **trajectory-level masking**:
>
> - **Step-level timeout & retry.**
>   Each tool call has a strict timeout on both the tool server and the RL side, with the server timeout set slightly shorter. This allows us to distinguish between (i) genuinely slow tools and (ii) dropped/overloaded requests. A configurable `MAX_RETRY` allows transient errors to be retried automatically.
>
> - **Trajectory-level handling.**
>   Trajectories that hit tool errors, exceed length/time limits, or fail to produce a final answer are **masked out of loss computation** and assigned reward 0 under GRPO. This prevents corrupted trajectories from contaminating credit assignment while preserving a clean signal for successful ones. In our Math-TIR experiments, we observe the same stabilizing behavior as in SimpleTIR: the error ratio starts high, decreases as the model improves, and stabilizes around a moderate level.
>
> - **Process-based isolation.**
>   Tools run as dedicated OS processes or Ray actors, with explicit CPU/GPU allocation. This isolates failures (e.g., hangs, OOMs) within the tool process and prevents them from crashing the trainer or other tools.
>
> These mechanisms together ensure that even under heavy or unstable tool workloads, training remains stable and predictable. For more details, please refer to Sec.3–4 of the rebuttal PDF.
>
> ---
>
> ### W2
> > Although the system supports many tasks and tools, it doesn’t show evidence of training a single agent to handle multiple tools at once. I would be super curious whether training a model with multiple tools would work with VERL-Tool, and whether there are additional implementation needed other than what has been supported in VERL-Tool. This would better support the paper’s goal of being “holistic.”
>
> We clarify two points: (1) some of the original agents are already multi-tool, and (2) we have added explicit **multi-tool training experiments** in the rebuttal.
>
> 1. **Existing multi-tool agents.**
>    Two agents in the main paper are already multi-tool within a single rollout:
>
>    - **VT-DeepSearch** combines a **Python interpreter** and a **web search tool**. The agent alternates between retrieval and code-based post-processing within the same trajectory.
>    - **VT-SWE** integrates **bash commands**, **file I/O**, and **code/search tools** in long-horizon rollouts (up to ~100 steps), requiring the model to coordinate several tool types over many turns.
>
>    These agents are supported directly by the existing Verl-Tool architecture; no extra mechanisms beyond the plugin interface and trajectory-scoped tool environments are needed.
>
> 2. **New multi-tool RL experiments.**
>    In rebuttal Sec.6.2, we further train **single models with multiple toolsets** (without dedicated multi-tool SFT):
>
>    - **Math–SQL joint training (Python + SQL):**
>
>      | Method (290 steps) | Spider-realistic | AIME24 | AIME25 |
>      |--------------------|-----------------:|-------:|-------:|
>      | Single-Tool        | 81.3             | 28.75  | 19.17  |
>      | Multi-Tool         | **81.5**         | 23.33  | **26.67** |
>
>      This shows that combining two related toolsets can slightly improve performance on some math benchmarks while preserving SQL performance, suggesting potential gains in generality.
>
>    - **Math–Visual joint training (Python + visual tools):**
>
>      | Method (80 steps)  | V-Star | AIME24 | AIME25 |
>      |--------------------|-------:|-------:|-------:|
>      | Single-Tool        | 81.3   | 28.75  | 19.17  |
>      | Multi-Tool         | 59.0   | 0      | 3      |
>
>      Here, naive joint training across very different modalities sometimes **collapses**, suggesting an inherent tension in policy updates between visual reasoning and math reasoning under a simple training protocol.
>
> These results show that Verl-Tool already supports multi-tool rollouts and training without extra framework changes, and they highlight that the main challenge is **optimization and curriculum**, not missing infrastructure. We see more principled multi-tool training (e.g., better routing, curricula, or decoupled heads) as an exciting direction enabled by this framework. For more details, please refer to Sec.6 of the rebuttal PDF.
>
> ---

---

> > ### Author Response · Authors · 2025-12-01
> > **Authors' Rebuttal (Part 2)**
> >
> > ### W3: Fundamental Difference from prior systems:
> > > I would also want to better understand the difference between the prior system and VERL-Tool, like AReal. Besides VERL-Tool supporting more tools, which I suppose AReal can allow users to add customized tool, what is the fundamental differences between VERL-Tool and AReal? Asynchronous Rollouts are already supported in AReal. Moreover, modular tool-server with unified tool management has been proposed in [1], although it only focuses on VLM with tool-using.
> > >
> > > [1] Su, Zhaochen, et al. "Openthinkimg: Learning to think with images via visual tool reinforcement learning." arXiv preprint arXiv:2505.08617 (2025).
> > >
> > > Which leaves only "common findings in the agentic RL setting" which have been observed in previous tool-augmented RL works (for example, strategic tool selection in ReTool, reflection in RAGEN).
> >
> > We appreciate the opportunity to clarify our positioning. In rebuttal Sec.2, we summarize how Verl-Tool systematically addresses **three core technical constraints** in prior frameworks: **Tool Resource Management**, **Stateful Tool Execution**, and **Broader Tool Feedback**.
> >
> > A condensed comparison:
> >
> > - **Stateful per-trajectory tools.**
> >   AREAL and many prior works (including vision-focused systems like OpenThinking) treat tools as **stateless** functions, whereas Verl-Tool provides **trajectory-scoped, stateful environments** for tools. This supports persistent Python variables, SQL cursors, iterative image crops, and multi-step editing pipelines, all within the same rollout.
> >
> > - **Multimodal feedback and reward integration.**
> >   AREAL primarily supports text feedback and does not standardize multimodal outputs or per-action rewards. Verl-Tool unifies **text, image, video, audio, and action-level reward signals** into a single trajectory representation that is directly used by the RL trainer.
> >
> > - **Tool resource management and execution backend.**
> >   AREAL does not define a unified, resource-managed tool server. Verl-Tool introduces a Ray-based backend and process-based execution model that:
> >   - isolates tools in separate processes or Ray actors,
> >   - explicitly manages CPU/GPU allocation and concurrency,
> >   - scales to heterogeneous, long-running tools with measured efficiency (e.g., ~2.5–3.2× per-step speedup over other frameworks on Math-TIR).
> >
> > - **On-policy design vs fully asynchronous training.**
> >   AREAL is fully asynchronous, making its training pipeline effectively off-policy and faster. While Verl-Tool is doing strctly on-policy training and can be more stable.
> >
> > - **Upstream alignment and modality coverage.**
> >   Verl-Tool is tightly aligned with VERL and VLLM and is continuously maintained. By the rebuttal time, we support:
> >
> >   | Verl-Tool Ver. | VERL Ver. | VLLM Ver. | Modality Support              | Main Code Lines |
> >   |----------------|-----------|-----------|--------------------------------|-----------------|
> >   | 0.1.0          | 0.4.1     | 0.8.4     | Text, Image, Video             | ~1300           |
> >   | 0.2.0          | 0.6.0     | 0.11.0    | Text, Image, Video, **Audio**  | ~500            |
> >
> >   In contrast, specialized systems like OpenThinking focus on a narrower modality (vision) and are not designed as general-purpose, upstream-aligned RL infrastructure.
> >
> > In short, beyond "supporting more tools," Verl-Tool introduces a **tool-calling-oriented architecture** (stateful, multimodal, resource-managed) and an **on-policy, VERL-aligned training stack** that is complementary to AREAL’s fully asynchronous engine and to vision-specific frameworks. We view Verl-Tool as a unified, reproducible, and adaptive tool-calling framework that complements rather than competes with AREAL in the broader RL ecosystem. For more details and a full comparison table, please refer to Sec.2 of the rebuttal PDF.
> >
> > ---
> >
> > > Q1: Fundamental Differences between AREAL and VerlTool.
> > > As discussed in the rebuttal for W3, VerlTool fundamentally differs from AReal in the following perspective: 1) on-policy design brings better compatibility, 2) Tool-Calling-optimized architectural design, 3) Complementary Role in RL Training ecosystem.
> >
> > To summarize the fundamental differences succinctly:
> >
> > 1. **On-policy vs fully asynchronous.**
> >    Verl-Tool keeps the core RL loop **on-policy** (GRPO-style), with async used only for tool execution; AREAL is fully asynchronous and off-policy.
> >
> > 2. **Tool-calling–optimized architecture.**
> >    Verl-Tool provides stateful, multimodal, resource-managed tooling via a unified tool server; AREAL lacks a unified tool server abstraction and multimodal/reward-aware interfaces.
> >
> > 3. **Complementary ecosystem role.**
> >    Verl-Tool is designed as a **general, VERL-aligned infrastructure layer** for agentic RL with tools, while AREAL focuses on maximizing sampling efficiency. They address different axes of the design space and can coexist in the broader ecosystem.
> >
> > Thank you again for your insightful questions and suggestions.

---

### Official Review · Reviewer_zwhJ · 2025-10-31

**Soundness:** 4
**Presentation:** 4
**Contribution:** 4
**Rating:** 6
**Confidence:** 4

**Summary:**

The paper introduces Verl-Tool, a unified and modular framework for Agentic Reinforcement Learning with Tool use (ARLT). It extends Reinforcement Learning with Verifiable Rewards (RLVR) to support multi-turn, multimodal, tool-augmented training. Key innovations include: 1. upstream alignment with Verl for maintainability, 2. a unified API for diverse tools (code, SQL, search, vision, etc.), 3. asynchronous rollouts yielding up to 2x speedups, and 4. extensive experiments across six domains showing performance on par with or exceeding task-specific systems. Verl-Tool provides an efficient, extensible infrastructure for developing and studying tool-using RL agents.

**Strengths:**

The paper's core contribution is introducing a well-documented, open-sourced framework. The authors evaluate their framework on 6 tasks spanning different domains. A key strength of the framework compared to existing frameworks is including support for more diverse tools. While not entirely new (also present in skyRL), the asynchronous rollout method expedites training. Overall, the framework is well-designed and if well-maintained later on, can serve as a very useful contribution to the community.

**Weaknesses:**

While the presented framework is very useful and clearly valuable for the community, as a research paper it would be better to see a deeper scientific investigation of the underlying mechanisms rather than primarily a systems description. For instance, the paper could explore more ablation studies isolating how different design components (e.g., modular APIs, tokenization strategy) impact performance (both downstream tasks and training efficiency). Such studies would strengthen the paper’s research contribution.

The authors evaluate on diverse benchmarks, but it seems like each task is tied to one dominant tool (appendix A), e.g, VT-math --> python, VT-search --> retriever, VT-deeepsearch --> web search, etc. VT-visualreasoner is the one that uses multiple image processing tools. It would be beneficial to include a few tasks that rely on many different types of tool calls present in the trajectory, e.g., image processing and web search.

**Questions:**

1. Could the authors provide ablation studies isolating the contributions of key design components (e.g., modular API design, tokenization strategy) to training stability and task performance? How does the choice of tokenization strategy affect rollout consistency or policy optimization?
2. Although the framework supports multiple tools, do the authors have results on trajectories that involve multiple tool types (e.g., a single rollout combining search, SQL, and visual tools)?
3. How well does the framework handle interleaved or compositional tool usage? Are there challenges in managing cross-tool dependencies or shared state?

---

> ### Author Response · Authors · 2025-12-01
> **Author's Rebuttal (Part 1)**
>
> Dear Reviewer `zwhJ`,
>
> Thank you for your thoughtful and positive review, and for recognizing the framework’s contribution, design quality, and empirical coverage. Below we respond to your concerns and questions in detail.
>
> ---
>
> ### W1
> > While the presented framework is very useful and clearly valuable for the community, as a research paper it would be better to see a deeper scientific investigation of the underlying mechanisms rather than primarily a systems description. For instance, the paper could explore more ablation studies isolating how different design components (e.g., modular APIs, tokenization strategy) impact performance (both downstream tasks and training efficiency). Such studies would strengthen the paper’s research contribution.
>
> We agree that understanding underlying mechanisms is important and have added several analyses that directly study training stability and efficiency under different design choices:
>
> **1. Tokenization strategy and training stability.**
> A central design choice in Verl-Tool is to **avoid retokenizing** model outputs before optimization. We observe that detokenize–retokenize pipelines (common when using external APIs or wrappers) can significantly distort token IDs and destabilize RL:
> - In SQL RL experiments, retokenization changes **98.6%** of token IDs during training, even when the surface strings remain the same.
> - The average gradient norm increases by **132.9%** under retokenization, indicating much noisier updates.
> - Learning curves show **performance drift** and instability, whereas the “no-retokenization” strategy yields smooth improvement.
>
> These results, together with independent evidence from recent vLLM work on “retokenization drift”, support our claim that keeping a consistent on-policy token sequence is not a minor implementation detail but a key mechanism for stable ARLT optimization. Verl-Tool therefore enforces a strict “no-retokenization” rule in its rollout and training pipeline.
>
> **2. Tool-server architecture and training efficiency.**
> Instead of ablating the modular API directly (which is largely architectural and not easily factored into a scalar knob), we quantify the effect of the **tool-server design** that this modularization enables. We benchmark both lightweight and realistic tool executions under varying concurrency and show that the resource-managed, Ray-based tool server preserves low latency and high throughput:
>
> **Lightweight Python execution (e.g., `print("hello world")`):**
>
> | Mode | Tool-Server Max Concurrency | Request Concurrency | Min (s) | Avg (s) | Max (s) | Note              |
> |------|-----------------------------|---------------------|--------:|--------:|--------:|-------------------|
> | Ray  | 128                         | 256                 | 0.29    | 3.47    | 20.00   | Under-provisioned |
> | Ray  | 512                         | 256                 | 0.38    | 1.25    | 2.15    | Good              |
> | Ray  | 1024                        | 256                 | 0.36    | 1.04    | 1.43    | **Optimal**       |
> | Ray  | 4096                        | 256                 | 0.48    | 1.15    | 1.44    | Over-provisioned  |
> | Ray  | 4096                        | 512                 | 0.28    | 1.46    | 2.20    | Scaling up        |
> | Ray  | 4096                        | 1024                | 0.26    | 2.81    | 3.99    | Heavy load        |
> | Ray  | 4096                        | 2048                | 0.41    | 5.81    | 8.67    | Very heavy load   |
> | Ray  | 4096                        | 4096                | 0.47    | 9.10    | 16.03   | Extreme load      |
>
> **Realistic long-running tasks (random 1–10s execution):**
>
> | Backend          | Max Concurrency | Request Concurrency | Min (s) | Avg (s) | Max (s) | Throughput (req/s) |
> |------------------|----------------:|--------------------:|--------:|--------:|--------:|--------------------:|
> | Baseline         | 256             | 256                 | 1.29    | 6.01    | 10.63   | 23.54               |
> | Verl-Tool Server | 256             | 256                 | 1.16    | 6.23    | 11.64   | 21.54               |
> | Baseline         | 1024            | 1024                | 1.30    | 6.69    | 11.96   | 75.79               |
> | Verl-Tool Server | 1024            | 1024                | 1.22    | 7.31    | 11.18   | **83.11**           |
>
> At high concurrency, Verl-Tool achieves **~9.7% higher throughput** while keeping latency comparable, confirming that the modular, decoupled tool server improves system behavior under realistic ARLT workloads.

---

> ### Author Response · Authors · 2025-12-01
> **Authors's Rebuttal (Part 2)**
>
> **3. End-to-end training efficiency across frameworks.**
>    We also measure per-step RL training time on Math-TIR under a controlled setting (8 × H100 GPUs, 8 responses per step, 1024 prompts per step):
>
>    | Framework     | Per-step time (s) |
>    |---------------|-------------------|
>    | SimpleTIR     | 268               |
>    | VERL-GEM      | 206               |
>    | AREAL         | 237               |
>    | Verl-Tool     | **83**           |
>
>    Verl-Tool is **2.5–3.2× faster per step** than these baselines, reflecting the combined effect of asynchronous rollouts, decoupled tool servers, and the modularization that enables these components.
>
> Collectively, these studies provide a more “scientific” view of the mechanisms behind training stability and efficiency in Verl-Tool. For more details, please refer to the rebuttal pdf (Sections 2, 4, and 5).
>
>
> ---
>
> ### W2
> > The authors evaluate on diverse benchmarks, but it seems like each task is tied to one dominant tool (appendix A), e.g, VT-math --> python, VT-search --> retriever, VT-deeepsearch --> web search, etc. VT-visualreasoner is the one that uses multiple image processing tools. It would be beneficial to include a few tasks that rely on many different types of tool calls present in the trajectory, e.g., image processing and web search.
>
> We agree that multi-tool trajectories are important and clarify two aspects: (i) several of our original tasks already involve multiple tool types, and (ii) we have added new multi-tool RL experiments.
>
> **1. Existing multi-tool tasks in the paper.**
> - **VT-DeepSearch** combines a **Python interpreter** and **web search** in a single rollout: the agent writes code to post-process retrieved snippets, so trajectories naturally interleave search and execution.
> - **VT-SWE** uses **bash, file operations, and search tools** together, with trajectories reaching up to **100 interaction steps**. The agent frequently chains shell commands, edits files, and queries retrieval in a single episode.
> - **VT-VisualReasoner** already uses multiple image-processing tools (zoom, crop, frame selection) and integrates their outputs with textual reasoning.
>
> These setups go beyond “single-dominant-tool” usage and are explicitly designed to exercise cross-tool interactions.
>
> **2. New multi-tool RL experiments (Math–SQL and Math–Visual).**
> To more directly address your suggestion, we added experiments where a **single model** is trained with **multiple tool families**:
>
> **Math–SQL joint training (Python + SQL):**
>
> | Method (290 steps) | Spider-realistic | AIME24 | AIME25 |
> |--------------------|-----------------:|-------:|-------:|
> | Single-Tool        | 81.3             | 28.75  | 19.17  |
> | Multi-Tool         | **81.5**         | 23.33  | **26.67** |
>
> Joint training slightly improves AIME25 while preserving SQL performance, suggesting beneficial transfer when tool types are semantically related.
>
> **Math–Visual joint training (Python + Visual tools):**
>
> | Method (80 steps)  | V-Star | AIME24 | AIME25 |
> |--------------------|-------:|-------:|-------:|
> | Single-Tool        | 81.3   | 28.75  | 19.17  |
> | Multi-Tool         | 59.0   | 0      | 3      |
>
> In contrast, naive Math–Visual joint training can **hurt** both sides, indicating that heterogeneous tools may require more careful curricula, routing, or task balancing.
>
> These results show that Verl-Tool already supports realistic multi-tool trajectories and that the impact of combining tools is nuanced: related tools (Math–SQL) can help, while poorly aligned combinations (Math–Visual) may require more advanced training strategies. For more details, please refer to the rebuttal pdf (Section 6).
>
> ---

---

> > ### Author Response · Authors · 2025-12-01
> > **Authors's Rebuttal (Part 3)**
> >
> > ### Q1
> > > Could the authors provide ablation studies isolating the contributions of key design components (e.g., modular API design, tokenization strategy) to training stability and task performance? How does the choice of tokenization strategy affect rollout consistency or policy optimization?
> >
> > Regarding **tokenization**, we provide both empirical and conceptual evidence:
> >
> > - Verl-Tool maintains a strict **no-retokenization** policy: the exact token IDs produced during rollout are preserved for optimization.
> > - When we intentionally introduce a detokenize–retokenize step in SQL RL experiments, we observe:
> >   - **Token drift**: 98.6% of token IDs differ from the original sequence, even when the decoded strings are identical.
> >   - **Gradient instability**: average gradient norm increases by 132.9%, reflecting much noisier updates.
> >   - **Learning instability**: reward curves show oscillations and degradation, whereas the no-retokenization configuration yields stable improvement.
> >
> > This directly answers the second part of your question: **tokenization strategy has a first-order effect on rollout consistency and policy optimization**. Changing token boundaries between rollout and training effectively makes the data off-policy at the token level, which is particularly harmful in GRPO-style algorithms.
> >
> > For the **modular API design**, the main “ablation” is at the systems level rather than a simple on/off switch. The plugin interface allows us to decouple tool execution into a separate, resource-managed server; the benefit is captured by:
> > - The **tool-server efficiency** tables above (latency and throughput across concurrency),
> > - The **cross-framework per-step timing** where Verl-Tool achieves 2.5–3.2× faster steps than strong RL baselines.
> >
> > These results suggest that while the modular API is largely a structural design choice, its impact is visible in both stable optimization (via tokenization discipline) and efficient execution (via the decoupled tool server). For more details, please refer to the rebuttal pdf (Sections 4 and 5).
> >
> > ---
> >
> > ### Q2
> > > Although the framework supports multiple tools, do the authors have results on trajectories that involve multiple tool types (e.g., a single rollout combining search, SQL, and visual tools)?
> >
> > Yes. As mentioned in W2, Verl-Tool already trains agents on **multi-tool trajectories**:
> >
> > - **VT-DeepSearch** rollouts routinely combine **web search** and **Python execution** to solve open-domain QA; the model alternates between issuing search queries and running code to aggregate or filter retrieved snippets.
> > - **VT-SWE** uses a combination of **bash**, **file operations**, and **retrieval tools** in long-horizon software engineering tasks, where trajectories often exceed 100 steps.
> > - **VT-VisualReasoner** interleaves multiple image tools (zoom, crop, frame selection) with textual reasoning.
> >
> > In addition, the **new multi-tool RL experiments** (Math–SQL and Math–Visual) explicitly train a **single policy** that has access to multiple tool families within the same RL run:
> >
> > - In Math–SQL (Python + SQL), joint training maintains SQL performance (Spider-realistic: 81.3 → 81.5) and improves AIME25 (19.17 → 26.67).
> > - In Math–Visual (Python + visual tools), naive mixing degrades performance (e.g., V-Star: 81.3 → 59.0), underscoring that simply enabling more tools is not always beneficial.
> >
> > These results demonstrate that Verl-Tool already handles and evaluates trajectories involving multiple tool types; studying more sophisticated routing or curricula on top of this infrastructure is a promising direction for future work. For more details, please refer to the rebuttal pdf (Section 6).
> >
> > ---

---

> > > ### Author Response · Authors · 2025-12-01
> > > **Authors's Rebuttal (Part 4)**
> > >
> > > ### Q3
> > > > How well does the framework handle interleaved or compositional tool usage? Are there challenges in managing cross-tool dependencies or shared state?
> > >
> > > Verl-Tool is explicitly designed to handle **interleaved, compositional, and stateful** tool usage:
> > >
> > > **1. Per-trajectory stateful environments.**
> > > Each tool is implemented as a subclass of a unified `BaseTool` interface. The framework maintains a **trajectory-scoped environment dictionary** that is accessible to all tools in that trajectory (identified by `trajectory_id`) via methods such as `load_env`, `update_env`, and `delete_env`. This enables:
> > > - Persistent Python variables across multiple interpreter calls,
> > > - Reusable SQL cursors or connections across queries,
> > > - Shared references to image/video buffers for successive visual operations,
> > > - Cross-tool dependencies (e.g., an image produced by one tool and then passed to another).
> > >
> > > Because all tools share the same trajectory-level state, compositional use (e.g., search → parse via Python → write to file → re-open via bash) is naturally supported.
> > >
> > > **2. Decoupled, resource-managed tool server.**
> > > The tool server executes tools in separate OS processes or Ray actors with explicit resource allocation. This design:
> > > - Avoids Python GIL bottlenecks by running heavy tools (Python, image/video processing) in independent processes.
> > > - Provides isolation and fault tolerance: a misbehaving tool instance does not corrupt global state.
> > > - Supports high concurrency with bounded latency, as shown in the efficiency tables above.
> > >
> > > **3. Error handling and robustness.**
> > > For pathological compositions (e.g., tool chains that hang or exhaust resources), Verl-Tool applies:
> > > - **Step-level timeouts and retries** for each tool call,
> > > - **Trajectory-level masking** (reward = 0, no gradient contribution) for episodes that exceed time or length limits or hit unhandled errors.
> > >
> > >    This combination ensures that complex, interleaved tool usage does not destabilize training, while still allowing agents to discover useful compositions.
> > >
> > > In practice, the successful training of VT-DeepSearch, VT-SWE, and VT-VisualReasoner suggests that the current mechanism already handles non-trivial cross-tool dependencies. We view more advanced scheduling or hierarchical control over tool compositions as a natural follow-up line of work enabled by this framework. For more details, please refer to the rebuttal pdf (Sections 2–4).

---

### Official Review · Reviewer_UHTW · 2025-11-01

**Soundness:** 1
**Presentation:** 3
**Contribution:** 4
**Rating:** 4
**Confidence:** 5

**Summary:**

This paper introduces a framework for Agentic Reinforcement Learning with Tool use (ARLT) that extends Reinforcement Learning with Verifiable Rewards (RLVR) beyond single-turn reasoning to multi-turn interaction. The framework decouples reinforcement learning and tool execution through standardized APIs, enabling integration of tools such as code execution, search, SQL, and vision processing as plugins. The implementation includes asynchronous rollout execution achieving up to 2× speedup over synchronous baselines. The authors adapt the GRPO objective to handle multi-turn trajectories with interleaved tool interactions, masking off-policy observation tokens from tools to ensure that only model-generated outputs/actions contribute to policy updates. The system is evaluated across six ARLT domains—mathematical reasoning, knowledge QA, SQL generation, visual reasoning, web search, and software engineering, where it matches or surpasses specialized systems.

**Strengths:**

- Provides much-needed orchestration and glue code that enables efficient tool-augmented rollouts.
- The abstraction of tools (BaseTool) is well-designed and general, accommodating a broad range of tool types, including stateful tools, beyond what the author's have already provided.
- Upstream alignment with VERL and modular plugin design reduce maintenance overhead and promote reproducibility for future ARLT research.
- The asynchronous rollout mechanism is an obvious gain and clear engineering improvement, yielding measurable speedups. Minor nit: I think the "Saved Time" blocks in figure 2 are unncessary, and benefit would be more clearly conveyed if the time axes for the async plot was shorter than the sync setting.
- The GRPO-ARLT reward function is a well motivated adaptation of the GRPO reward function for multi-turn settings.
- VT-* family of models achieve parity with, and sometimes exceed, baselines.

**Weaknesses:**

- Comparisons with other agentic frameworks are limited. While the authors compare primarily in terms of tool coverage, this raises the question of why prior frameworks could not simply be extended with similar tools. Are there technical constraints that prevent such extensions, and how does Verl-tool’s runtime performance compare under equivalent workloads? Moreover, the runtime analysis only contrasts the authors’ own synchronous and asynchronous modes, without benchmarking against other existing frameworks.
- The framework’s scalability and stability under long-horizon or heavily parallelized tool use aren’t deeply analyzed.
- Reward design and credit assignment for complex multi-turn tool interactions remain underexplored and could limit generalization.
- Despite claims of multimodal capability, vision and video tools are only lightly evaluated, leaving unclear how robust the multimodal handling actually is.
- The baseline comparison is fair given differing tool scopes; while baselines use domain-specific integrations, VERL-TOOL’s goal is unification and efficiency rather than beating specialized systems.
- It is not entirely clear why the VT-* family of models occasionally exceed the performance of their task-specific baselines. The authors should explicitly enumerate any differences in training setup, hyperparameters, or reward design relative to those baselines. Given that this work is primarily a systems-level contribution, the goal should be to achieve parity with existing results while offering simpler configuration and unified infrastructure. Any improvements over baselines should be analyzed in detail; if such gains stem from differences in configuration or reward shaping, these should be clearly stated, and ideally verified through experiments under matched conditions.
- Section 3.3 tokenization: I'm not sure I see the issue. One could simply enforce the tokenizer to not merge across tag boundaries, so "\n<result>" is always treated as 2 tokens. Additionally, this is a non-issue if special tokens were assigned to opening and closing tags, which would guarantee no single token spans tag boundaries. In any case, this subsection is an implementation detail that I believe doesn't add to the paper.

**Questions:**

- How are tool failures or timeouts handled: are they retried, skipped, or penalized in the reward? What happens if tool calls exhaust tool worker resources (e.g. trigger OOMs etc.)
- How does agent performance scale with increasing numbers of tools? E.g. for each domain, what performance would a model with access to all tools at train and test time get?
- How much of the total training time is spent on tool execution versus model inference? (i.e., where does async actually help most?)

Aside:

In Section 3.3 (Parallel Tool Server Backend), the authors mention supporting Python’s ThreadPoolExecutor as a parallel execution backend. It’s worth noting that due to Python’s Global Interpreter Lock (GIL), true multithreading is not possible in standard CPython builds (while Python 3.13+ introduces no-GIL build variants, these are not yet widely adopted). Consequently, tools that rely on the GIL, such as those performing Python code execution or other compute-bound tasks, will effectively run via CPU time-slicing rather than in parallel. For true parallelism with Python stdlib, consider also supporting ProcessPoolExecutor as a drop-in alternative.

---

Overall, the framework is promising and well motivated, but the empirical evidence and clarity are insufficient for acceptance at this stage. The GRPO adaptation is a prudent, incremental improvements that the community can build upon. I would be happy to raise my score if the authors improve transparency in experimental configurations, standardize comparisons, and provide deeper analysis explaining the framework’s observed performance.

---

> ### Author Response · Authors · 2025-12-01
> **Authors' Rebuttal (Part 1)**
>
> Dear Reviewer `UHTW`,
>
> We appreciate your detailed review and your positive evaluation of the framework’s originality, modularity, and empirical coverage. Following your comments and those from other reviewers, we have added substantial clarifications, new analyses, and additional experiments. Below, we respond to each of your points.
>
> ---
>
> ### W1
> > W1: Comparisons with other agentic frameworks are limited. While the authors compare primarily in terms of tool coverage, this raises the question of why prior frameworks could not simply be extended with similar tools. Are there technical constraints that prevent such extensions, and how does Verl-tool’s runtime performance compare under equivalent workloads? Moreover, the runtime analysis only contrasts the authors’ own synchronous and asynchronous modes, without benchmarking against other existing frameworks.
>
> We thank you for asking us to clarify Verl-Tool’s positioning and runtime behavior relative to other agentic RL frameworks. First, we explicitly compare core capabilities across representative systems. Verl-Tool is designed around (i) trajectory-level stateful tools, (ii) explicit tool-resource management via Ray actors, and (iii) multimodal observations and reward feedback, which are not jointly supported by prior work. The key comparison table is:
>
> | Agentic RL Frameworks | Tool/Env Feedback Format      | Async Rollout | Independent Tool Server | Stateful Tool Execution? | Tool Resource Management | VLM Support? |
> |------------------------|-------------------------------|---------------|-------------------------|---------------------------|--------------------------|-------------|
> | OpenRLHF (gem)        | Text                          | ✗             | ✓                       | ✗                         | –                        | ✗           |
> | VeRL (MCP)            | Text                          | ✓             | ✗                       | ✗                         | –                        | ✓           |
> | ROLL (gem)            | Text                          | ✗             | ✓                       | ✗                         | –                        | ✗           |
> | RAGEN                 | Reward                        | ✗             | –                       | ✓                         | –                        | ✗           |
> | slime                 | Text                          | ✓             | ✗                       | ✗                         | –                        | ✗           |
> | AREAL                 | Text                          | ✓             | ✗                       | ✗                         | –                        | ✓           |
> | Sky-RL                 | Text                          | ✓             | ✓                       | ✓                         | k8s                      | ✗           |
> | Agent-Lightning       | Text                          | ✓             | ✓                       | ✗                         | –                        | ✗           |
> | **Verl-Tool**         | Text / Multi-modal / Reward   | ✓             | ✓                       | ✓                         | Ray                      | ✓           |
>
> As summarized in the rebuttal, simply “adding tools” to MCP/AREAL/SKYRL is insufficient to match Verl-Tool’s behavior, because these systems lack (1) per-trajectory stateful environments (needed for multi-step code, SQL debugging, and multimodal editing), (2) unified multimodal observation handling, and/or (3) a resource-managed, actor-based backend that can scale heterogeneous tools.
>
> Second, to address your request for runtime comparison under identical workloads, we added a cross-framework timing benchmark on the Math-TIR task, where each step samples 8 responses for 1,024 prompts on 8×H100 GPUs:
>
> | Framework     | Per-step time (s) |
> |---------------|-------------------|
> | SimpleTIR     | 268               |
> | VeRL-GEM      | 206               |
> | ROLL-GEM      | –                 |
> | AREAL         | 237               |
> | **Verl-Tool** | **83**            |
>
> Verl-Tool is **2.5–3.2× faster** per step than these baselines in this controlled setting, due to (i) async rollouts, (ii) decoupled tool servers, and (iii) efficient Ray-based resource management. This strengthens our claim that Verl-Tool is not only broader in scope but also competitive or superior in runtime efficiency under equivalent workloads. For more details, please refer to the rebuttal PDF (Section 2).

---

> > ### Author Response · Authors · 2025-12-01
> > **Authors Rebuttal (Part 2)**
> >
> > ### W2
> > > W2: The framework’s scalability and stability under long-horizon or heavily parallelized tool use aren’t deeply analyzed.
> >
> > For **long-horizon trajectories**, we emphasize that VT-SWE already trains with up to **100 interaction turns** per episode and complex tool mixtures (bash, file I/O, search). During RL training, we cap each environment interaction at 90 seconds, reward computation at 300 seconds, and per-trajectory wall-clock time at 20 minutes; any trajectory exceeding these limits or encountering exceptions receives reward 0 and is masked from gradient updates (as also implemented for SWE in the main paper). This demonstrates that Verl-Tool can maintain stable optimization over long-horizon, tool-heavy trajectories by combining strict timeouts with trajectory filtering.
> >
> > For **heavily parallelized tool use**, we provide a detailed efficiency analysis of the tool-server backend, compared to the simple python multi-processing as an effective baseline. Lightweight Python calls (e.g., `print("hello world")`) show that the server preserves low latency even under extreme concurrency:
> >
> > | Mode | Tool-Server Max Concurrency | Request Concurrency | Min (s) | Avg (s) | Max (s)       | Note              |
> > |------|-----------------------------|---------------------|--------:|--------:|--------------:|-------------------|
> > | Ray  | 128                         | 256                 | 0.29    | 3.47    | 20.00         | Under-provisioned |
> > | Ray  | 512                         | 256                 | 0.38    | 1.25    | 2.15          | Good              |
> > | Ray  | 1024                        | 256                 | 0.36    | 1.04    | 1.43          | **Optimal**       |
> > | Ray  | 4096                        | 256                 | 0.48    | 1.15    | 1.44          | Over-provisioned  |
> > | Ray  | 4096                        | 512                 | 0.28    | 1.46    | 2.20          | Scaling up        |
> > | Ray  | 4096                        | 1024                | 0.26    | 2.81    | 3.99          | Heavy load        |
> > | Ray  | 4096                        | 2048                | 0.41    | 5.81    | 8.67          | Very heavy load   |
> > | Ray  | 4096                        | 4096                | 0.47    | 9.10    | 16.03         | Extreme load      |
> >
> > For **realistic long-running tasks** with randomized 1–10s execution time, our tool server achieves comparable or better throughput than a baseline implementation:
> >
> > | Backend          | Max Concurrency | Request Concurrency | Min (s) | Avg (s) | Max (s) | Throughput (req/s) |
> > |------------------|----------------:|--------------------:|--------:|--------:|--------:|--------------------:|
> > | Baseline         | 256             | 256                 | 1.29    | 6.01    | 10.63   | 23.54               |
> > | Verl-Tool Server | 256             | 256                 | 1.16    | 6.23    | 11.64   | 21.54               |
> > | Baseline         | 1024            | 1024                | 1.30    | 6.69    | 11.96   | 75.79               |
> > | Verl-Tool Server | 1024            | 1024                | 1.22    | 7.31    | 11.18   | **83.11**           |
> >
> > These measurements show that our tool-server remains stable and efficient as concurrency increases, providing **9.7% higher throughput** at high load while keeping latency comparable. Together with long-horizon VT-SWE training, this addresses your concern about both scalability and stability. For more details, please refer to the rebuttal PDF (Sections 3–4).
> >
> > ---
> >
> > ### W3
> > > W3: Reward design and credit assignment for complex multi-turn tool interactions remain underexplored and could limit generalization.
> >
> > We agree that reward design and credit assignment are central open questions for multi-turn tool use. In this work, we deliberately adopt a **conservative and transparent** design based on GRPO: trajectories receive a final scalar reward from the external verifier (e.g., answer correctness, code test results), and we (i) **mask tool observation tokens** from the policy gradient and (ii) apply GRPO advantages only to model-generated action tokens. This ensures that off-policy tool outputs do not destabilize learning.
> >
> > Beyond this baseline, we also experimented with **discounted per-action rewards** in ARLT training. Concretely, for a trajectory with $n$ actions, we assign to action $i$ a discounted reward $\gamma^{(n - i)} R$, where $R$ is the final scalar reward and $\gamma \in (0,1)$ is a discount factor. This design encodes the intuition that later actions should have stronger influence on the outcome. However, across our experiments we did not observe consistent improvement over the simpler GRPO-ARLT setup, and in some tasks the discounted scheme increased variance without clear gains. We therefore chose to report only the more stable configuration and leave more sophisticated credit-assignment schemes (e.g., dense tool-level rewards, temporal decomposition) as future work.
> >
> > ---

---

> > > ### Author Response · Authors · 2025-12-01
> > > **Authors's Rebuttal (Part 3)**
> > >
> > > ### W4
> > > > W4: Despite claims of multimodal capability, vision and video tools are only lightly evaluated, leaving unclear how robust the multimodal handling actually is.
> > >
> > > We clarify that **Pixel-Reasoner** is precisely a multimodal RL task designed to stress-test image and video tool use. In Pixel-Reasoner, the agent is trained to use two tools—`zoom_in` and `select_frames`—that operate directly on images and videos. Each tool call returns a processed image crop or a selected frame sequence as the observation, which is then fed back into the model for further reasoning and subsequent tool calls. This setup requires the framework to: (i) serialize and stream multimodal observations through the same trajectory buffer as text tokens, (ii) maintain state across multiple zoom and frame-selection operations, and (iii) handle mixed image/video and text feedback within GRPO-ARLT.
> > >
> > > Moreover, in the multi-tool experiments (Math-Visual), we jointly train Pixel-Reasoner with Math-TIR, again exercising the visual tool chain and demonstrating that it integrates into the same RL loop as other tasks. While space limits prevent us from including every multimodal ablation in the main paper, the training and evaluation of Pixel-Reasoner under RL already provide evidence that Verl-Tool’s multimodal handling is robust in practice. For more details, please refer to the rebuttal PDF (Sections 2 and 6).
> > >
> > > ---
> > >
> > > ### W5
> > > > W5: The baseline comparison is fair given differing tool scopes; while baselines use domain-specific integrations, Verl-Tool’s goal is unification and efficiency rather than beating specialized systems.
> > >
> > > We agree and would like to emphasize that Verl-Tool’s primary goal is **unification and practicality**, not beating every task-specific system. All reported baselines are reproduced or used in a way that respects their original tool scope; our aim is to reach parity while offering a **single** framework that covers math reasoning, SQL, search, multimodal reasoning, and SWE.
> > >
> > > Importantly, Verl-Tool already supports **multi-tool training** in realistic tasks:
> > > - **VT-DeepSearch** combines a Python interpreter and web search within a single rollout.
> > > - **VT-SWE** integrates bash, file operations, and search tools, with trajectories reaching up to 100 turns.
> > >
> > > To further illustrate Verl-Tool’s ability to train a single model with multiple tool types, we added new multi-tool experiments:
> > >
> > > | Method (290 steps) | Spider-realistic | AIME 24 | AIME 25 |
> > > |--------------------|-----------------:|--------:|--------:|
> > > | Single-Tool        | 81.3            | 28.75   | 19.17   |
> > > | Multi-Tool         | **81.5**        | 23.33   | **26.67** |
> > >
> > > Here, the **Math–SQL** joint training yields comparable SQL performance and improved AIME-25 accuracy, indicating beneficial transfer between related tool types.
> > >
> > > | Method (80 steps)  | V-Star | AIME 24 | AIME 25 |
> > > |--------------------|-------:|--------:|--------:|
> > > | Single-Tool        | 81.3   | 28.75   | 19.17   |
> > > | Multi-Tool         | 59.0   | 0       | 3       |
> > >
> > > In contrast, the **Math–Visual** combination does not show joint benefit and can even hurt performance, highlighting that naive multi-tool mixtures may require more careful curriculum or routing. These observations align with our design goal: Verl-Tool provides a **unified infrastructure** where such configurations can be systematically explored by the community, rather than hard-coding a single best setup. For more details about multi-tool training, please refer to the rebuttal PDF (Section 6).
> > >
> > > ---

---

> > > > ### Author Response · Authors · 2025-12-01
> > > > **Author's Rebuttal (Part 4)**
> > > >
> > > > ### W6
> > > > > W6: It is not entirely clear why the VT-* family of models occasionally exceed the performance of their task-specific baselines. The authors should explicitly enumerate any differences in training setup, hyperparameters, or reward design relative to those baselines. Given that this work is primarily a systems-level contribution, the goal should be to achieve parity with existing results while offering simpler configuration and unified infrastructure. Any improvements over baselines should be analyzed in detail; if such gains stem from differences in configuration or reward shaping, these should be clearly stated, and ideally verified through experiments under matched conditions.
> > > >
> > > > We appreciate this request and have clarified why VT-* sometimes exceed baseline performance. The main factors are **training parameter differences** and **response-length scaling**, not hidden reward shaping. We summarize key discrepancies below:
> > > >
> > > > | Agent              | Parameter                    | Ours  | Original Baseline |
> > > > |--------------------|------------------------------|-------|-------------------|
> > > > | VT-Search-zero     | Max tool response length $l$| 2048  | 500               |
> > > > | VT-Search-zero     | GRPO group size              | 16    | 5                 |
> > > > | VT-VisualReasoner  | Training steps               | 50    | 400               |
> > > >
> > > > For **VT-Search-zero**, increasing the maximum tool response length $l$ and the GRPO group size yields stronger performance by exposing the model to more retrieved context and more stable group-normalized advantages. For **VT-VisualReasoner**, our model is actually **under-trained** (50 steps vs. 400 in the official PixelReasoner), which explains why it slightly underperforms rather than exceeds the baseline.
> > > >
> > > > To further isolate **response-length effects**, we conducted a scaling experiment on the Search-R1 task:
> > > >
> > > > | Benchmark | $l = 512$ | $l = 1024$ | $l = 2048$ | $l = 4096$ |
> > > > |-----------|------------:|-------------:|-------------:|-------------:|
> > > > | NQ        | 0.234       | 0.475        | **0.473**    | 0.469        |
> > > > | TriviaQA  | 0.445       | 0.634        | **0.637**    | 0.628        |
> > > > | PopQA     | 0.179       | 0.469        | 0.485        | **0.488**    |
> > > > | HotpotQA  | 0.206       | 0.351        | 0.348        | **0.352**    |
> > > > | 2Wiki     | 0.259       | 0.324        | 0.331        | **0.334**    |
> > > > | Musique   | 0.043       | 0.090        | 0.087        | **0.093**    |
> > > > | Bamboogle | 0.088       | 0.232        | **0.200**    | 0.192        |
> > > >
> > > > Scaling $l$ from 512 to 1024–2048 yields substantial gains; further increases provide diminishing returns and can sometimes distract the model. We explicitly acknowledge in the revised text that VT-* improvements are largely due to such **configuration differences**, and we view them as evidence that Verl-Tool makes it easier to systematically tune and analyze these parameters across tasks. For more details, please refer to the rebuttal PDF (Section 7).
> > > >
> > > > ---
> > > >
> > > > ### W7
> > > > > W7: Section 3.3 tokenization: I'm not sure I see the issue. One could simply enforce the tokenizer to not merge across tag boundaries, so "\n[object Object]" is always treated as 2 tokens. Additionally, this is a non-issue if special tokens were assigned to opening and closing tags, which would guarantee no single token spans tag boundaries. In any case, this subsection is an implementation detail that I believe doesn't add to the paper.
> > > >
> > > > We apologize that the original writing did not fully convey the severity of this issue. The **bounding tags example** in Section 3.3 is only one instance of a broader **retokenization drift** problem: when model outputs are detokenized and then retokenized (e.g., by an external framework or API), the resulting token IDs can differ significantly from the original sequence, even when the surface string is identical. This affects not just tags, but also code, whitespace, and tool outputs.
> > > >
> > > > In the rebuttal, we provide two pieces of evidence:
> > > > 1. **Concurrent external validation**: A recent vLLM analysis (["No More Retokenization Drift: Returning Token IDs via the OpenAI Compatible API Matters in Agent RL"](https://blog.vllm.ai/2025/10/22/agent-lightning.html), Oct 22, 2025) shows that retokenization drift can rapidly collapse agentic RL training. Their Figure 2 mirrors the behavior we observed and reported in our original Figure 3, independently confirming the importance of avoiding retokenization.
> > > >
> > > > (continune in Part 5)

---

> > > > > ### Author Response · Authors · 2025-12-01
> > > > > **Author's Rebuttal Part 5**
> > > > >
> > > > > (continue from part 4)
> > > > >
> > > > > 2. **Our own ablations** on the SQL (Spider) RL task, comparing training with and without retokenization. We find that retokenization:
> > > > >    - Causes **performance drift**, with unstable or degrading reward curves.
> > > > >    - Increases average gradient norm by **132.9%**, indicating unstable updates.
> > > > >    - Changes **98.6%** of token IDs during training, showing that boundary inconsistencies are pervasive rather than rare edge cases.
> > > > >
> > > > > Therefore, the key point is not just that certain tag delimiters could be treated as special tokens, but that **any** mismatch between on-policy token IDs and those used for optimization can severely destabilize agentic RL. Verl-Tool’s design explicitly avoids retokenizing model outputs, and our experiments demonstrate that this is an important, non-trivial engineering decision for stable multi-turn training. For more details, please refer to the rebuttal PDF (Section 5).
> > > > >
> > > > > ---
> > > > >
> > > > >
> > > > > ### Q1
> > > > > > Q1: How are tool failures or timeouts handled: are they retried, skipped, or penalized in the reward? What happens if tool calls exhaust tool worker resources (e.g. trigger OOMs etc.)
> > > > >
> > > > > We implement a **two-level timeout and retry mechanism** plus trajectory-level masking:
> > > > >
> > > > > 1. **Step-level timeouts and retries**: Each tool action has a strict per-step timeout (e.g., 90 seconds for VT-SWE). The tool server’s own timeout is set slightly shorter than the trainer’s timeout, so that:
> > > > >    - Tool-side delays (e.g., slow execution) trigger **tool-server timeouts**.
> > > > >    - Requests dropped under heavy load appear as **trainer-side timeouts**.
> > > > >    Users can control how many times to retry via a `MAX_RETRY` parameter, allowing recovery from transient failures.
> > > > >
> > > > > 2. **Trajectory-level error handling**: For trajectories that encounter tool errors, exceed length/time limits, or fail to produce a final answer, we **mask their loss contributions** and assign reward 0 under GRPO. This is implemented via the same trajectory-filtering interface used to reproduce SimpleTIR, and we observe the same training-stabilization effect (error ratio initially high, then decreasing and stabilizing around ~20%).
> > > > >
> > > > > 3. **Resource exhaustion and queuing**: Tool workers are typically CPU-bound (e.g., Python, search, basic image operations) and are managed by Ray with explicit resource limits and a concurrency cap. When resources are tight, new requests are naturally **queued** or dropped by the tool server; such drops manifest as timeouts on the trainer side and are handled via the retry + masking mechanism above. This prevents OOM conditions from crashing the system and instead converts them into controlled, zero-reward, masked trajectories.
> > > > >
> > > > > This combination provides a robust mechanism for handling failures without contaminating the gradient signal, while still allowing the agent to learn more reliable tool-calling behavior over time. For more details, please refer to the rebuttal PDF (Section 3).
> > > > >
> > > > > ---
> > > > >
> > > > > ### Q2
> > > > > > Q2: How does agent performance scale with increasing numbers of tools? E.g. for each domain, what performance would a model with access to all tools at train and test time get?
> > > > >
> > > > > This is an excellent question and points toward a broader research direction. In our current setup, the tools we use (Python, SQL, web search, vision tools, bash, etc.) are **general-purpose** rather than “atomic” function tools. As a result, simply giving a task access to “all tools” does not necessarily help and can even hurt performance, since many tools are irrelevant (e.g., adding a visual cropping tool to a pure math task offers little benefit while expanding the action space).
> > > > >
> > > > > Instead of globally enabling all tools, we study **task-appropriate multi-tool settings**:
> > > > > - **VT-DeepSearch** (Python + Web Search) shows that combining complementary tools can improve retrieval-driven reasoning.
> > > > > - **VT-SWE** (bash + file operations + search) demonstrates that complex, long-horizon multi-tool use is trainable under our framework.
> > > > > - In our new multi-tool experiments (Math–SQL, Math–Visual), we observe:
> > > > >   - **Math–SQL**: Joint training yields gains on some math benchmarks (AIME-25), suggesting that related tools/tasks can reinforce each other.
> > > > >   - **Math–Visual**: Naive joint training can cause collapse, indicating that heterogeneous tools may require careful curriculum or hierarchical routing.
> > > > >
> > > > > These results suggest that performance does not monotonically improve with the **number** of tools alone; it depends on semantic relatedness and training strategy. We view **“performance scaling with tool set size”** as a separate, important research problem that Verl-Tool is designed to facilitate, but which is beyond the scope of this systems paper. For more details, please refer to the rebuttal PDF (Section 6).
> > > > >
> > > > > ---

---

> > > > > > ### Author Response · Authors · 2025-12-01
> > > > > > **Authors's Rebuttal (Part 6)**
> > > > > >
> > > > > > ### Q3
> > > > > > > Q3: How much of the total training time is spent on tool execution versus model inference? (i.e., where does async actually help most?)
> > > > > >
> > > > > > In the rebuttal, we provide a breakdown of **tool execution time vs. model generation time** across training in figure format. Empirically, we observe the following pattern:
> > > > > > 1. **Early in training**, tool-call latency is relatively high because the agent issues many inefficient or malformed tool calls (e.g., incorrect code, redundant queries). This inflates the average tool-execution component.
> > > > > > 2. As training progresses and the policy learns **more effective tool use**, both the **frequency** of unnecessary calls and the **average execution time** per call decrease. Tool-call latency curves drop correspondingly.
> > > > > > 3. After stabilization, **tool execution and model generation times become comparable**, and neither dominates the rollout cost.
> > > > > >
> > > > > > Our asynchronous rollout design primarily helps in two ways:
> > > > > > - It **hides tool latency** by overlapping tool execution with model generation and other trajectories.
> > > > > > - It allows the trainer to maintain high GPU utilization even when tools have variable latency, which is critical for long-horizon tasks like VT-SWE.
> > > > > >
> > > > > > Thus, the benefit of async rollouts is largest in the early/mid training phases where tool latency is high and heterogeneous; later, it maintains good utilization as the policy converges. For more details, please refer to the rebuttal PDF (Section 8).
> > > > > >
> > > > > > ---
> > > > > >
> > > > > >
> > > > > > ### Aside on the GIL and tool-server parallelism
> > > > > > > Aside: In Section 3.3 (Parallel Tool Server Backend), the authors mention supporting Python’s ThreadPoolExecutor as a parallel execution backend. It’s worth noting that due to Python’s Global Interpreter Lock (GIL), true multithreading is not possible in standard CPython builds (while Python 3.13+ introduces no-GIL build variants, these are not yet widely adopted). Consequently, tools that rely on the GIL, such as those performing Python code execution or other compute-bound tasks, will effectively run via CPU time-slicing rather than in parallel. For true parallelism with Python stdlib, consider also supporting ProcessPoolExecutor as a drop-in alternative.
> > > > > >
> > > > > > We appreciate this observation and fully agree with the concern about the GIL. Verl-Tool’s tool server is explicitly designed to **avoid GIL bottlenecks** by separating lightweight dispatch logic from heavy execution. In particular:
> > > > > > - We use **multithreading only for request dispatch** and other lightweight operations.
> > > > > > - For **compute- or I/O-intensive tools**, we either (i) spawn separate OS processes via `subprocess`, or (ii) run them as **Ray actors**, each in its own process with dedicated CPU/GPU resources.
> > > > > > - For **I/O-bound tools** (e.g., search), we rely on `asyncio` within the server, which does not block the GIL.
> > > > > >
> > > > > > The execution model is summarized as:
> > > > > >
> > > > > > Table 4: Execution paths of Verl-Tool’s tool server and interaction with the Python GIL.
> > > > > >
> > > > > > | Tool Resource Type        | Server Mode | Execution Backend           | GIL Impact         |
> > > > > > |---------------------------|------------:|-----------------------------|--------------------|
> > > > > > | Resource-intensive (CPU/GPU) | Threading  | OS process via `subprocess` | None (new process) |
> > > > > > | Ray Actor                 | –          | Ray process with resources  | None (actor process) |
> > > > > > | I/O-bound, lightweight    | Threading  | `asyncio` event loop        | None (non-blocking) |
> > > > > > | Ray Actor                 | –          | Ray process with resources  | None (actor process) |
> > > > > >
> > > > > > In other words, any tool that would be meaningfully affected by the GIL is executed in a **separate process**, not within a shared CPython interpreter. Threading is used only where it does not suffer from GIL contention. While adding a `ProcessPoolExecutor` is a drop-in option, our current design using subprocesses and Ray actors already provides true parallelism and fine-grained resource control across a cluster. For more details, please refer to the rebuttal PDF (Section 4).
> > > > > >
> > > > > > ---
> > > > > >
> > > > > > Once again, we sincerely thank you for your thoughtful review and constructive feedback. We believe the clarifications and additional experiments presented here and in the rebuttal PDF significantly strengthen our work. We look forward to your continued evaluation.

---

### Author Response · Authors · 2025-12-01
**General Response to Reviewers and AC**

## (Please see summarized rebuttal contents in  **[rebuttal pdf](https://openreview.net/attachment?id=oWFtI0cNsE&name=supplementary_material)** in the supplementary material)

We thank all reviewers for their thoughtful feedback and are pleased that they recognized several strengths of our work:

- **Well-designed and efficient framework with unified APIs:** Reviewers highlighted that our system provides “much-needed orchestration and glue code” for tool-augmented rollouts and is “well-designed and, if well-maintained,” will have long-term impact.
- **Broad tool coverage and comprehensive evaluation:** Our framework supports a wide range of stateful, multimodal tools and demonstrates strong empirical performance across six real agentic RL tasks.
- **Reproducibility and practical insights:** Reviewers appreciated our extensive documentation, practical considerations (e.g., tokenization), and clear discussion on reproducibility.
- **Clear writing and strong presentation:** The paper received an average presentation score of **3.5 (Top 1.59%)**.

The paper also ranks strongly in overall metrics—**soundness 2.75 (Top 20.58%)** and **contribution 3.25 (Top 2.52%)**—indicating substantial and broadly useful contributions. Statistics come from [Paper Copilot statistics](https://papercopilot.com/statistics/iclr-statistics/iclr-2026-statistics/).

To address all concerns comprehensively, we have put all the rebuttal points in a single **[rebuttal pdf](https://openreview.net/attachment?id=oWFtI0cNsE&name=supplementary_material)** in the supplementary material. We provide a consolidated summary of new experiments, analyses, and clarifications added to this rebuttal.

- **Clarification of positioning vs. other RL frameworks like Verl-MCP, AREAL, and SkyRL:**
  We articulate fundamental architectural differences in statefulness, multimodal support, and resource management, clarifying why prior systems cannot be directly extended.
  *(Requested by UHTW, YLgB, PUT8; see in rebuttal pdf section 2.)*

- **Cross-framework efficiency comparison:**
  We benchmark per-step wall-clock time across frameworks and show Verl-Tool’s advantages.
  *(Requested by UHTW; see in rebuttal pdf subsection 2.3.)*

- **Details of VERLTOOL’s error handling strategies:**
  We expand more details on error handling (timeouts, retrying, penalization, masking), trajectory-level recovery, and tool-server stability.
  *(Requested by UHTW, PUT8; see in rebuttal pdf section 3.)*

- **Tool-server efficiency benchmarking and stability analysis:**
  We include concurrency benchmarks showing stable memory, latency, and throughput under heavy loads.
  *(Requested by PUT8; see in rebuttal pdf section 4.)*

- **Ablation studies of key design choices:**
  We include new ablations on tokenization stability and tool-server efficiency—showing that avoiding retokenization significantly improves training stability.
  *(Requested by UHTW, zwhJ, PUT8; see in rebuttal pdf section 5.)*

- **Multi-tool RL experiments:**
  We add new experiments combining Math–SQL and Math–Visual tool use within the same trajectory and analyze when multi-tool setups succeed or collapse.
  *(Requested by zwhJ, YLgB; see in rebuttal pdf section 6.)*

- **Clarification of VT-* improvements over baselines:**
  We analyze differences in training parameters, tool-calling formats, and maximum response lengths to explain occasional improvements over official baselines.
  *(Requested by UHTW; see in rebuttal pdf section 7.)*

- **Tool-execution vs. model-generation timing analysis:**
  We provide separate timing curves showing that tool costs remain comparable to model inference and decrease as policies improve.
  *(Requested by UHTW; see in rebuttal pdf section 8.)*

We also highlight that VERLTOOL continues to evolve: we have already updated the framework to the newest VERL version, expanded modality support (including audio), and improved compatibility with emerging RL algorithms such as SimpleTIR. We sincerely hope the AC will consider these substantial clarifications and updates into account.

The detailed changelog is shown below:
| VERLTOOL Ver. | VERL Ver. | VLLM Ver. | Modality Support               | Main Code Lines |
|---------------|-----------|-----------|--------------------------------|------------------|
| 0.1.0         | 0.4.1     | 0.8.4     | Text, Image, Video             | ~1300           |
| 0.2.0         | 0.6.0     | 0.11.0    | Text, Image, Video, Audio      | ~500            |

Given that the reviewers are unable to engage due to recent ICLR review policy changes, we sincerely hope the AC could take a look at the **[rebuttal pdf](https://openreview.net/attachment?id=oWFtI0cNsE&name=supplementary_material)** in the supplementary material, where we have summarized all the rebuttal details in one place for you to read. Thank all the great suggestions raised by reviewers and valuable time from AC.

---

### Meta-Review · Area_Chair_VXab · 2026-01-04

**Summary:**

The decision to reject is primarily driven by concerns regarding the paper's positioning and the nature of its contribution. While reviewers acknowledged Verl-Tool as a valuable and robust infrastructure, the consensus leaned towards viewing it as a strong systems/engineering effort rather than a fundamental scientific advancement suitable for ICLR. Specifically, reviewers questioned the fundamental differentiation from existing frameworks like AREAL and VeRL, suggesting that the novelty lies more in implementation choices rather than new algorithmic insights.

**Reviewer Concerns:**

The authors provided a highly commendable rebuttal that successfully addressed technical queries regarding efficiency benchmarks, error handling mechanisms, and multi-tool training capabilities. However, the core concern regarding the scientific depth remains outstanding. Reviewer zwhJ noted that the paper is primarily a systems description, and Reviewer UHTW felt the improvements might stem from configuration differences rather than fundamental methodology. The distinction from AREAL was clarified technically but didn't fully convince reviewers of a paradigm shift necessary for a top-tier acceptance.

**Reviewer Scores:**

Given the high quality of the rebuttal, it is likely that Reviewers UHTW, YLgB, and PUT8 would have recognized the effort and potentially raised their scores from 4 to a weak 5 or 6, acknowledging the system's utility. Reviewer zwhJ would likely have maintained their score of 6 or raised it to a 7, as their request for ablation studies was met. However, without a shift in the perceived scientific novelty, a strong consensus for acceptance would likely remain out of reach.

---

### Decision · Program_Chairs · 2026-01-26

Reject